# Structural basis for the inactivation of cytosolic DNA sensing by the vaccinia virus

Angel Rivera-Calzada [1,4], Raquel Arribas-Bosacoma [2,3,4], Alba Ruiz-Ramos[1], Paloma Escudero-Bravo[1], Jasminka Boskovic [1], Rafael Fernandez-Leiro [1], Antony W. Oliver [2], Laurence H. Pearl [2,3] ✉ & Oscar Llorca [1] ✉

Detection of cytosolic DNA is a central element of the innate immunity system against viral infection. The Ku heterodimer, a component of the NHEJ pathway of DNA repair in the nucleus, functions as DNA sensor that detects dsDNA of viruses that replicate in the cytoplasm. Vaccinia virus expresses two proteins, C4 and C16, that inactivate DNA sensing and enhance virulence. The structural basis for this is unknown. Here we determine the structure of the C16 – Ku complex using cryoEM. Ku binds dsDNA by a preformed ring but C16 sterically blocks this access route, abrogating binding to a dsDNA end and its insertion into DNA-PK, thereby averting signalling into the downstream innate immunity system. C4 replicates these activities using a domain with 54% identity to C16. Our results reveal how vaccinia virus subverts the capacity of Ku to recognize viral DNA.

In response to infection by pathogens such as bacteria and viruses, hosts have evolved mechanisms to detect pathogen-associated molecular patterns (PAMPs), molecular motifs conserved within a class of pathogen. In the case of viruses, detection of viral nucleic acids is an essential part of this innate immune response, and human cells have acquired several mechanisms to detect DNA in the cytoplasm[1,2]. These cytoplasmic DNA sensors include proteins such as cyclic guanosine monophosphate-AMP synthase (cGAS), DHX36 helicase, MRE11, and DNA-dependent protein kinase (DNA-PK)[1–3], which also functions in the nucleus in both V(D)J recombination and as a core component of the non-homologous end joining (NHEJ) system for DNA double-strand break (DSB) repair[4–8].

DNA-PK was identified as a pattern recognition receptor that binds cytoplasmic DNA and activates innate immunity[9,10]. The DNA-PK holoenzyme consists of the DNA-dependent protein kinase catalytic subunit (DNA-PKcs) and the Ku70-Ku80 heterodimer (named simply as Ku hereafter)[4–6,8] and both components of the holoenzyme participate in cytoplasmic DNA sensing and innate immunity[9–14].

Inevitably, pathogens have developed countermeasures against the host DNA sensors that attempt to block or delay the host immune response and allow the proliferation and spread of the disease. A number of viruses, including adenoviruses, Epstein-Barr virus, herpes simplex virus 1, human papillomavirus, hepatitis B virus, Kaposi sarcoma-associated herpesvirus and notably vaccinia virus (VACV)[15–17] achieve this through interacting with components of the NHEJ pathway.

VACV is a prototypical member of Poxviridae family, which replicates their dsDNA genomes in the host cell cytoplasm. VACV strain Western Reserve (WR) expresses two proteins, C4 and C16[15,16], that counteract the pattern recognition function of the DNA-PK holoenzyme[11,12]. Both of these VACV proteins have been shown to bind to Ku[11,12], which provides the primary high-affinity recognition of a DNA end[4–6,8], but not to DNA-PKcs. Binding of C4 and C16 to Ku abrogates its capacity to bind DNA and DNA-PKcs, resulting in inhibition of the innate immune response and enhanced virulence. Virus lacking either C4 or C16 have reduced virulence compared to wild-type, and this effect is stronger when both proteins are removed. VACV employs other strategies to inactivate DNA sensing besides C4/C16, including the targeting of cGAS and cGAMP and encoding for inhibitors of IRF3 or NF-kB signalling[17]. VACV poxin is a nuclease that helps

[1]Spanish National Cancer Research Centre (CNIO), Melchor Fernández Almagro 3, 28029 Madrid, Spain. [2]Genome Damage and Stability Centre, School of Life Sciences, University of Sussex, Falmer, Brighton BN1 9RQ, UK. [3]Division of Structural Biology, Institute of Cancer Research, Chester Beatty Laboratories, 237 Fulham Road, London SW1E 6BT, UK. [4]These authors contributed equally: Angel Rivera-Calzada, Raquel Arribas-Bosacoma. ✉e-mail: laurence.pearl@sussex.ac.uk; ollorca@cnio.es

evade innate immunity by cleaving 2′,3′-cGAMP and preventing the activation of the receptor stimulator of interferon genes (STING)[18].

C16 comprises 331 amino acids and a C-terminal fragment comprising residues 157–331 is sufficient to form a complex with Ku and disrupt the binding of Ku to DNA and DNA-PKcs[11]. However, the N-terminal half of C16 (residues 1–214) is dispensable for the interaction of C16 with Ku but this domain binds the human oxygen sensing enzyme prolyl-hydroxylase domain containing protein (PHD)2 and induces a hypoxic response[19]. C4 comprises 316 amino acids and shares around 42% identity overall with C16, with the highest similarity in the C-terminal half, which, as in C16, contains the binding region for Ku and it is sufficient to inhibit DNA sensing[12]. Neither C4 nor C16 bind DNA directly. Homologues of C16 and/or C4 can be identified in the genomes of other orthopoxviruses, including the clinically important variola (VARV)−the causative agent of smallpox, and monkeypox (MPXV), which has recently broken out of its endemic location with cases reported in multiple non-African locations in 2022.

Here, we have determined the structure of VACV-C16 in complex with the Ku subunit of DNA-PK by cryoEM, which, together with the study of C4 and the C4–Ku complex both biochemically and by cryoEM, reveals the mechanism by which the viral proteins disrupt the ability of Ku to recognize viral DNA. Both proteins function as dimers and share a C-terminal domain with more than 50% sequence identity that binds Ku. One subunit of the C-terminal domain interacts exclusively with the Ku80 subunit at one end of the central channel of the Ku heterodimer, placing the other subunit in position to insert an acidic plug into the mouth of the Ku ring, thereby sterically blocking the access to any DNA. We also describe how structural differences in the N-terminal domains of C4 and C16 and their distinct capacity to interact with other proteins could account for some of their functional differences.

## Results

### C16 and C4 both prevent DNA binding by Ku

To verify that both C16 and C4 were able to inhibit DNA binding to Ku in vitro, we set up an electrophoretic mobility shift assay (EMSA) in which the migration of a fluorescently labelled dsDNA molecule is retarded when it forms a tight complex with a protein present in the assay. As expected, we observed a robust shift in DNA mobility when purified Ku was added to the system (Fig. 1a) consistent with the formation of a Ku·DNA complex. Separately, we purified C4 and C16 (see "Methods") and found that neither protein, either individually or in combination caused any change in DNA mobility (Fig. 1b) suggesting that they do not directly interact with DNA. When titrated into the assay along with the Ku complex, both C4 and C16 completely abrogate DNA binding by Ku when present at a concentration equimolar to the DNA, indicating direct competition for binding (Fig. 1c). However, we observed no synergistic effect when C4 and C16 were present together, suggesting that they act through a similar mechanism. In addition, both C4 and C16 can out-compete DNA when the Ku-DNA complex is pre-assembled before adding C4 or C16 (Fig. 1d).

### C16 and Ku assemble a complex

For structural studies, C16 from the vaccinia virus was produced as a recombinant protein in E. coli and purified to homogeneity by affinity chromatography. C16 eluted from size exclusion chromatography (SEC) as two peaks corresponding to different oligomeric forms of the protein (Supplementary Fig. 1a), which could also be detected by SDS-PAGE when the mix of C16 and the loading buffer was not boiled (Supplementary Fig. 1b). Mass photometry confirmed that recombinant C16 assembles dimers, tetramers as well as larger oligomeric forms (Supplementary Fig. 1c). Human Ku was also expressed and purified, mixed with an excess of C16 and applied to a size exclusion column as before (Fig. 2a). Ku alone eluted closed to the 158 kDa molecular weight standard whereas Ku and C16 formed a stable complex, detected as a peak eluting faster than Ku and C16 and that

only appeared when both proteins were incubated together (Fig. 2a). SDS-PAGE of the peak fraction confirmed the presence of the C16 – Ku complex (Fig. 2b).

### CryoEM and flexibility of the C16–Ku complex

The C16–Ku complex was vitrified and analysed by cryoEM (Supplementary Fig. 2a, Supplementary Tables 1, 2). 2D averaging of the cryoEM images revealed some views that could be readily assigned as Ku with a protruding density corresponding to C16 (Fig. 2c). The most distal densities appeared blurred, indicating some flexibility. This was also apparent during 3D classification of the cryoEM data and thus several volumes could be reconstructed where the most distal N-terminal densities were flexibly attached to the rest of the structure (Supplementary Fig. 3). After modelling the structure of the complex (see later), it is apparent that the ordered segment corresponds to the C-terminal domain of C16 (C16-C hereafter) which engages with Ku, and also forms a dimer (Supplementary Fig. 3b). The more flexible domain corresponds to the N-terminal half of C16. 3D variance analysis using cryoSPARC[20] confirmed both flexible attachment of C16 to Ku and movements of the C16 N-terminal domains, but in addition revealed that binding of C16 induces a significant deformation of the Ku ring (Supplementary Movies 1, 2 and 3). Accordingly, local resolution estimates indicated that in these maps, resolution was higher for the regions corresponding to Ku (around 2.5 Å) and C16-C (around 2.9-3.4 Å) (Supplementary Fig. 4a). The more flexible domain corresponds to the N-terminal half of C16 (C16-N hereafter), and its density was smeared out by averaging during image processing (Supplementary Fig. 2b, Supplementary Fig. 4a). We addressed this by excluding the flexible regions of C16 during refinement and by combining several different strategies (see Methods) (Supplementary Fig. 2b–d). Out of 579,038 initial particles for the C16–Ku complex, 170,770 resulted in a map for the C16-C–Ku complex where, although the estimated average resolution was similar to the full complex, structural features were better defined (Supplementary Fig. 4b). Local refinement of C16-N resulted in a map obtained from 81,353 particles and that reached an estimated 3.47 Å average resolution (Supplementary Fig. 4c). A map for the full C16–Ku complex was then generated by merging the two sub-volumes and selecting a single position for C16-N (Fig. 2d).

### Structure of C16

In the absence of previous structural information, a prediction from AlphaFold[21] was used as starting point for modelling the atomic structure of the C16 (Supplementary Table 3). The amino acid sequence for both copies of C16 could be readily traced into the well-ordered features of the cryoEM map, from the genetically defined N-terminal methionine to the genetically defined C-terminus at Lys331, with the exception of a couple of more poorly ordered loops. The structure of C16 is comprised of two distinct domains connected by a short linker (Fig. 3a); a three-layer β-β-α sandwich domain comprising residues from the N-terminus to Asn161, and which has structural homology to 2-oxo-glutarate iron-dependent oxidases such as the hypoxia-inducible factor prolyl hydroxylase PHD2[22], plus a highly curved 5-stranded anti-parallel β-sheet domain with single α-helices packed against the concave and convex faces from Asp180 to the C-terminus, but with no significant structural homologues[23] (Fig. 3a). The two domains of C16 each form an approximately C2 symmetric dimer linked together by a β-hairpin and loops that allow for conformational flexibility between them.

Although the N-terminal domain closely resembles PHD2 in structure, and retains the cleft that accommodates the iron-oxoglutarate in that protein family, the key residues that bind the cofactor in these oxidase enzymes, are not conserved in C16, suggesting that it does not have enzymatic activity (Fig. 3b). Dimerisation of the N-terminal domains involves antiparallel β-sheet interactions between residues Met1−Tyr4 on the edge strands of the central β-sheet

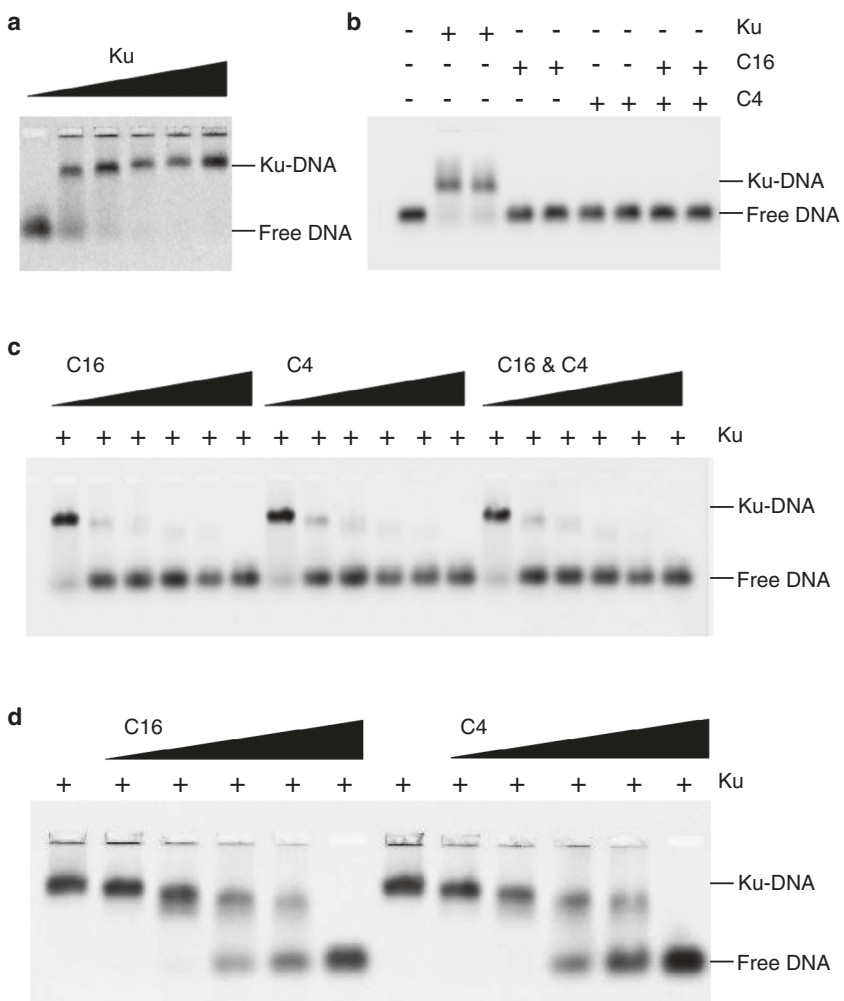

**Fig. 1 | C16 and C4 block DNA binding to Ku. a** Electrophoretic mobility shift assays (EMSA). A 5′ Cy3-labelled 19-bp duplex oligonucleotide (10 nM) was incubated with increasing amounts of recombinant Ku (0, 5, 10, 20, 30 and 40 nM) and the experiment visualised by the fluorescence of the Cy3 label attached to the DNA. The upper band indicates binding of DNA by the Ku heterodimer. **b** The same DNA substrate was incubated with recombinant Ku (20 and 30 nM), recombinant VACV C16 (100 and 150 nM), recombinant VACV C4 (100 and 150 nM) or C16 and C4 together (50 and 75 nM each). Neither of the VACV proteins bind DNA individually or in combination. **c** The same DNA substrate as in a was incubated with recombinant Ku (30 nM) in the presence of increasing amounts of C16 (5, 10, 20, 40, 80

and 120 nM), C4 (5, 10, 20, 40, 80 and 120 nM) or both proteins together (2.5, 5, 10, 20, 40 and 60 nM each). C4 and C16 are both effective in out-competing DNA binding to Ku at molar equivalence to the DNA. **d** The same experimental set-up as in c but pre-incubating the DNA substrate with 30 nM Ku at room temperature for 30 min prior to adding either C4 or C16 to the reaction mixture. The first column at the left of each experiment is loaded with the Ku-DNA complex before adding C4 or C16. Both C4 and C16 can out-compete DNA for binding to Ku. For all panels in Fig. 1, each EMSA was run in triplicate, and a representative gel is shown in the figure.

from each subunit, and between residues Gly123-Lys126 from the outer β-sheet from each subunit (Fig. 3b).

Interestingly, a piece of unattached density was evident, bound in a cleft formed at the interface of the two C16 molecules on the dyad axis of the N-terminal domains (Supplementary Fig. 5a). This protein density did not correspond to any segment of the C16 molecules in the homodimer, which were fully accounted for. Unfortunately, the resolution of the map did not permit unambiguous identification of the sequence of residues present in this extra density, but is expected to contain largely negatively charged amino acids since the walls of the cleft are intensely positively charged, formed by lysines 115, 117, 126, 140, 142 and 143 from each of the N-terminal domains (Supplementary Fig. 5b). This finding suggests that the cleft is likely to serve as a region for protein-protein interaction. A purely speculative but interesting possibility is that the unattached peptide corresponds to another C16 molecule, since C16 has the propensity to form larger oligomers after purification (Supplementary Fig. 1). In support of this possibility, we

found very distinctive 2D averages which could be interpreted as oligomers of C16 (Supplementary Fig. 5c). However, we were unable to reconstruct any volume with sufficient resolution that could be interpreted using these images, and we think this could be due to a lack of some views and/or the conformational heterogeneity. In a few images, density of a similar shape was found to be attached to C16 in the C16–Ku complex suggesting that this oligomerization could also take place within the context of the interaction with Ku (Supplementary Fig. 5d). Notably, the lysines involved in forming this cleft in C16 are not conserved in C4, which does not show the same propensity to oligomerisation (Supplementary Fig. 5b).

In the C-terminal domain, the core of the dimer interface involves resides 303–307 from the edge strand of the central β-sheet (Fig. 3c) of each subunit. The hydrophobic face of the side chain of Tyr305 from one chain, packs against the side chain of Val307 from the other chain, while its phenol oxygen along with the amine side chain of Lys321 interact with the carboxyl side chain of Asp259. Adjacent to this

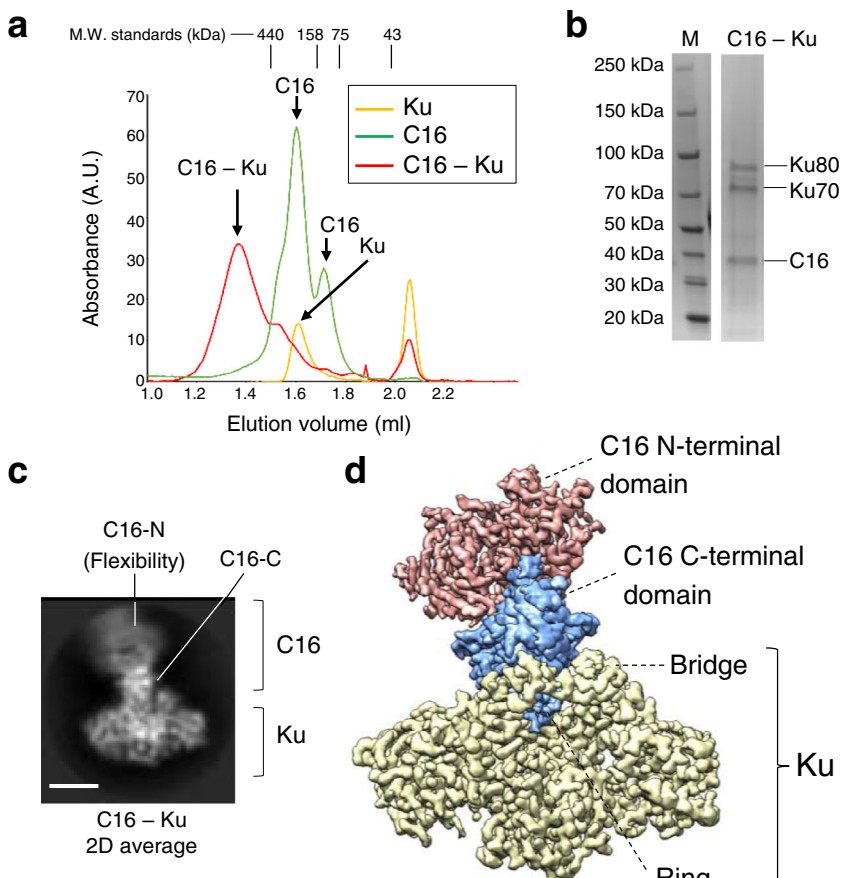

**Fig. 2 | Purification and cryoEM of the C16–Ku complex. a** SEC of C16, Ku and the mixture of both proteins. The elution volume for the peaks of standard proteins Ferritin (440 kDa), Aldolase (158 kDa), Conalbumin (75 kDa) and Ovalbumin (43 kDa) are indicated on top of the chromatogram. M.W. stands for Molecular Weight. **b** Right: SDS-PAGE of C16–Ku complex. Left: Molecular Weight (M) markers run in the same SDS-PAGE. The experiment shown in a and b was repeated three times with similar results. Source data are provided as a Source Data file. **c** One representative 2D average of the cryoEM images obtained for C16 –Ku, showing the flexibility of the more distal densities in C16. Scale bar represents 5 nm. C16-C and C16-N stand for the C- and N-terminal domains of C16 respectively. **d** CryoEM map of the C16 –Ku complex built as a composite merging the map of the C16-C– Ku and the C16-N (see Supplementary Fig. 2). Ku is shown in yellow, C16 N-terminal domains in red and C16 C-terminal domains in blue.

interaction, the phenyl ring of Phe303 makes a π-π stack with the guanidinium head group of Arg261 on the other chain, which is hydrogen bonded to the peptide backbone connecting Phe303 and Asp304 (Fig. 3c). The structure suggests that dimerisation of the C-terminal domains happens in the absence of the N-terminal domains, and accordingly we find that C16-C forms dimers on its own using SEC (Supplementary Fig. 5e). Since C16 and C4 show highest similarity in their C-terminal half, this suggests that C4 will also form dimers.

We evaluated if AlphaFold (AF)[21] and its version for protein-complex prediction[24] succeed in predicting the structure of C16. For this, each domain was first considered separately since they are flexibly connected. Root-mean-square deviation (RMSD) between AF predictions and the experimental structures were all under 1 Å, indicating that the structure of each domain is well predicted (Supplementary Fig. 6a). However, predictions fail entirely in determining the structure of the C16 dimer interface for both domains, at least for the conformation we find in complex with Ku and when we use the entire C16 sequence as an input for prediction (Supplementary Fig. 6b).

**Structure of the C16–Ku complex**
We then analysed the interaction between the C16 C-terminal domain and Ku. Here, the cryoEM map for the C16-C–Ku complex shows clear volumes corresponding to the Ku heterodimer, and the presence of two copies of the C-terminal domains from the vaccinia C16 protein to form a dimer without ambiguity (Fig. 4a). The C16 dimer binds as a single unit at one end of the central channel of the Ku heterodimer interacting exclusively with the Ku80 subunit (Fig. 4a). The two C16 subunits are stacked on top of each other so that the lower C16 interacts with the side and top rim of the core β-barrel domain of Ku80 that forms the base of the Ku DNA-binding channel, while the upper C16 is held over the top of the β-barrel and directed towards the large hole defined by the bridge structure that sits over the channel and forms part of the Ku heterodimer interface[25]. The analysis of the conformational flexibility of the C16–Ku complex also revealed that the formation of the complex still allows for a certain swinging movement of C16, which can result in some deformation of the bridge forming the DNA binding ring (Supplementary Movies 1, 2 and 3 and Supplementary Fig. 7a).

The lower C16 has a substantially hydrophobic interface with Ku80, with side chains of Tyr197, Leu198, Phe246, Leu252, Phe253, Thr312, Pro314 from C16 forming a curved hydrophobic groove into which the side chains of Val 2, Ile245, His246 and Pro248 of Ku80 protrude (Fig. 4b, c). This is supported by polar interactions of Arg368 and Asp369 of Ku80 with Ser247, Glu248 and Glu251 of C16, Lys399 and Arg400 of Ku80 with Asp194 and Asp195 of C16, and an interaction of the phenol OH of Tyr397 of Ku80 with the aromatic ring of Tyr197 of C16. The majority of these interacting residues are also conserved in C4 (Supplementary Fig. 5b).

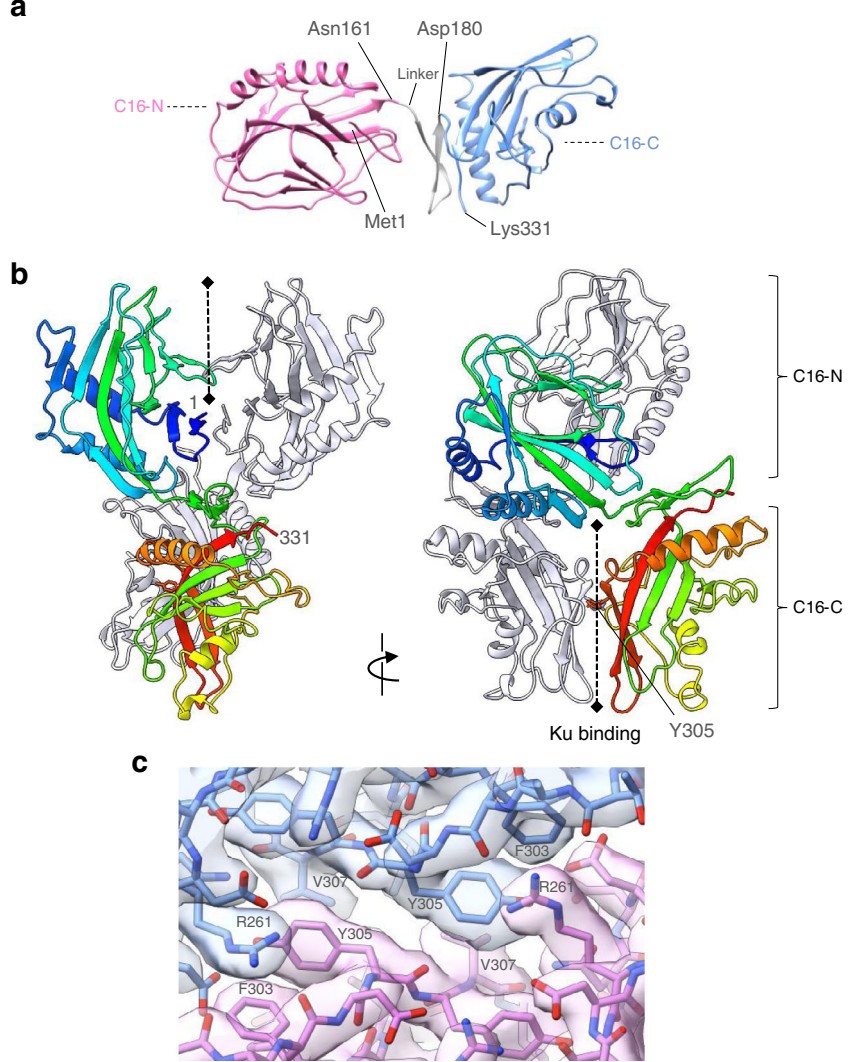

**Fig. 3 | Structure of vaccinia virus C16 protein. a** Boundaries between the N-terminal (pink colour) and C-terminal domain (light blue colour), of C16 represented as ribbons. The two domains are connected by a flexible linker. **b** Ribbon diagram of C16 dimer rainbow coloured (N-blue → C-red). The position of the first and last residue is indicated. C16 N-terminal domain has structural homology to PHD2[22] and contains a three-layer β-β-α sandwich domain. The C-terminal domain of vaccinia C16 has an α-β-α sandwich architecture consisting of a highly curved 5-stranded anti-parallel β-sheet domain with single α-helices packed against the concave and convex faces. It forms an approximately twofold symmetric dimer in its interaction with Ku, mediated primarily through interaction of the edge strand of the central β-sheet. The C-terminal domain of vaccinia C16 has no close structural homologue in the human genome[23]. C16-C and C16-N stand for the C- and N-terminal domains of C16 respectively. **c** Details of the C16 dimer interface. The atomic model is shown with the experimental cryoEM map.

The upper C16 subunit makes surprisingly few direct interactions with the Ku heterodimer apart from a polar interaction between Asp258 and Arg400 of Ku80, which also contacts the lower C16 molecule, and glancing van der Waals contacts made by Ile231 and Ser232 of C16 with Lys307 and Glu308 from Ku80 at the top corner of the bridge (Fig. 4d, e). However, supported by its dimer interaction with the lower C16, it projects an intensely negatively charged plug formed by a helix-turn-helix motif from residues 233-255 into the mouth of the bridge (Fig. 4e). The tip of this plug is formed by residues from D245 to E251 containing several acidic residues (Supplementary Fig. 7b) that insert into the Ku ring, which itself displays a positively charged surface (Fig. 4f).

A previous mutagenesis study suggested that a conserved Cys-Tyr-Cys motif−187−189 in C16 and 174−176 in C4 played a direct role in mediating interaction with Ku[12]. However, our structural data indicate that these residues are buried in the core of both subunits, so that the effect of their mutation on Ku interaction is likely due to gross

destabilisation of the C16/C4 C-terminal domain fold, rather than a more direct impact on the interface with Ku. Mapping C16 mutations and truncation into C16–Ku revealed suitable regions to disrupt the interaction between both proteins. Another study identified a fragment comprising residues 157–331 as sufficient for binding, but awkwardly when the protein was split in two fragments (residues 1–214 and 215–331) neither was able to interact with Ku[11]. Those findings can now be rationalized with the structure of the complex, since this truncation would fragment the C-terminal domain (Fig. 3a). This same study reported that an internal deletion of amino acids 277–281 severely reduced binding to Ku[11]. However, these residues locate in the C-terminal domain far from the region that contacts Ku, and they are not involved in the dimerization of C16-C. Thus, their effect on Ku binding must be indirect, perhaps by destabilising the conformation of C16-C.

We also analysed the heterogeneity in the cryoEM densities present at the opposite side of the Ku ring, which is not occupied

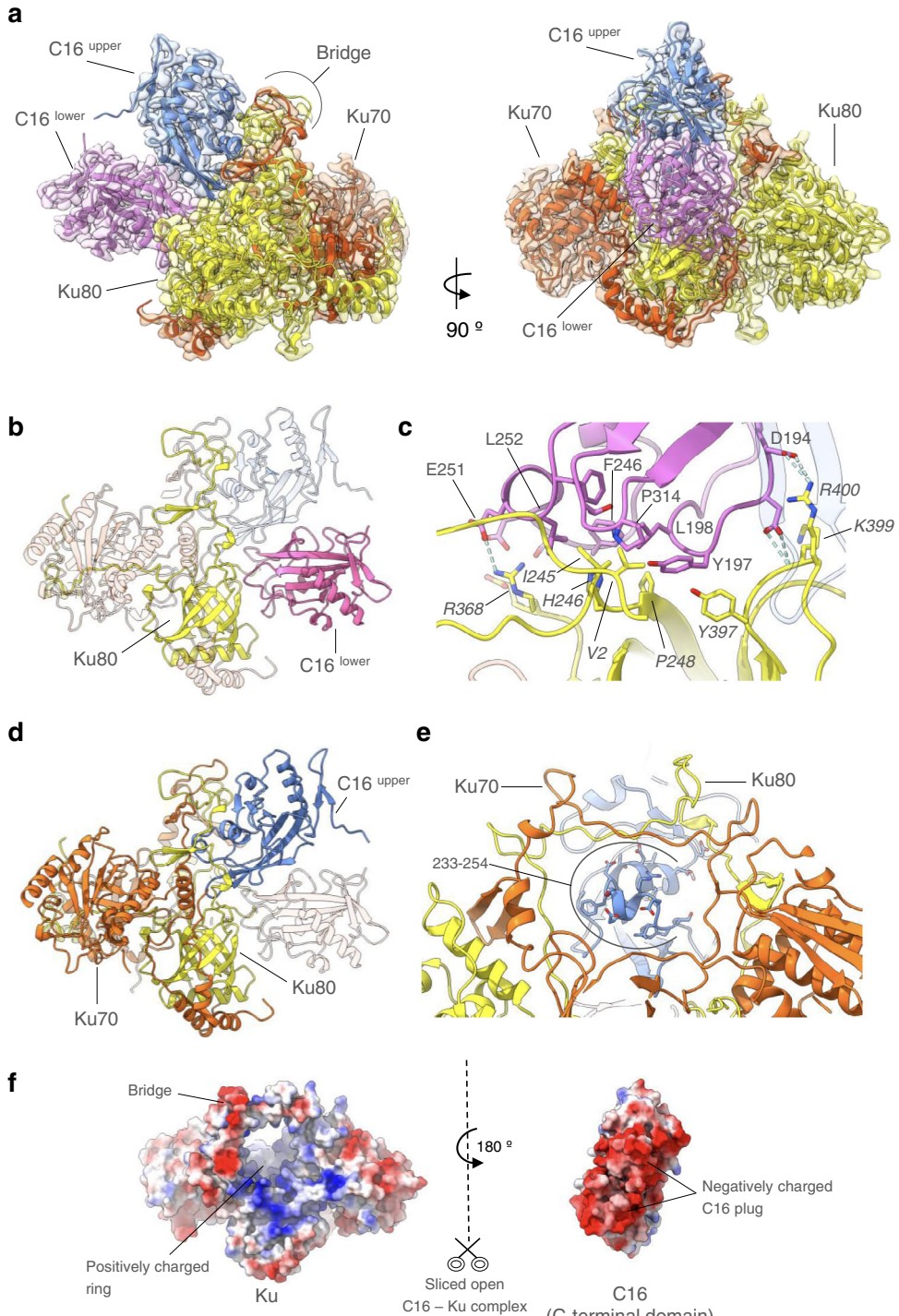

**Fig. 4 | Structure of the C16 – Ku complex. a** Orthogonal views of the atomic model of the C16–Ku complex in the C16-C–Ku experimental cryoEM map. The C16 dimer binds on one side of the Ku bridge structure, making contact only with the Ku80 subunit. **b** The lower C16 subunit interacts primarily with the rim of the core β-barrel of Ku80. **c** Details of the interaction of the lower C16 molecule with Ku80. The core of the interface is provided by packing of hydrophobic side chains of Ku80 into a curved hydrophobic groove on C16, reinforced by polar interactions at the periphery of the interface. Most of the residues involved from C16 are well conserved into the C4 paralogue. **d** The upper C16 makes few direct interactions with Ku, but is juxtaposed to the Ku bridge by its interaction with the lower C16. **e** Close-up of the projecting helix-turn-helix motif of the upper C16 which penetrates the space under the bridge. **f** Complementary electrostatic surfaces of Ku (left) and the C16 dimer (right) coloured blue +ve → red −ve. The view is the same as the right side of a, but the C16 dimer has been separated and rotated by 180° around the vertical to show the interacting surfaces.

by C16 (Supplementary Fig. 7c). We found that the ring shows various degrees of filling with a heterogenous density that in some classes had a clear continuity with the Ku70 chain. This density could correspond to the N-terminal region of Ku70, which has not been modelled in the available crystal structures, and which is enriched in negatively charged amino acids that could potentially interact with the positive charges at the ring in Ku.

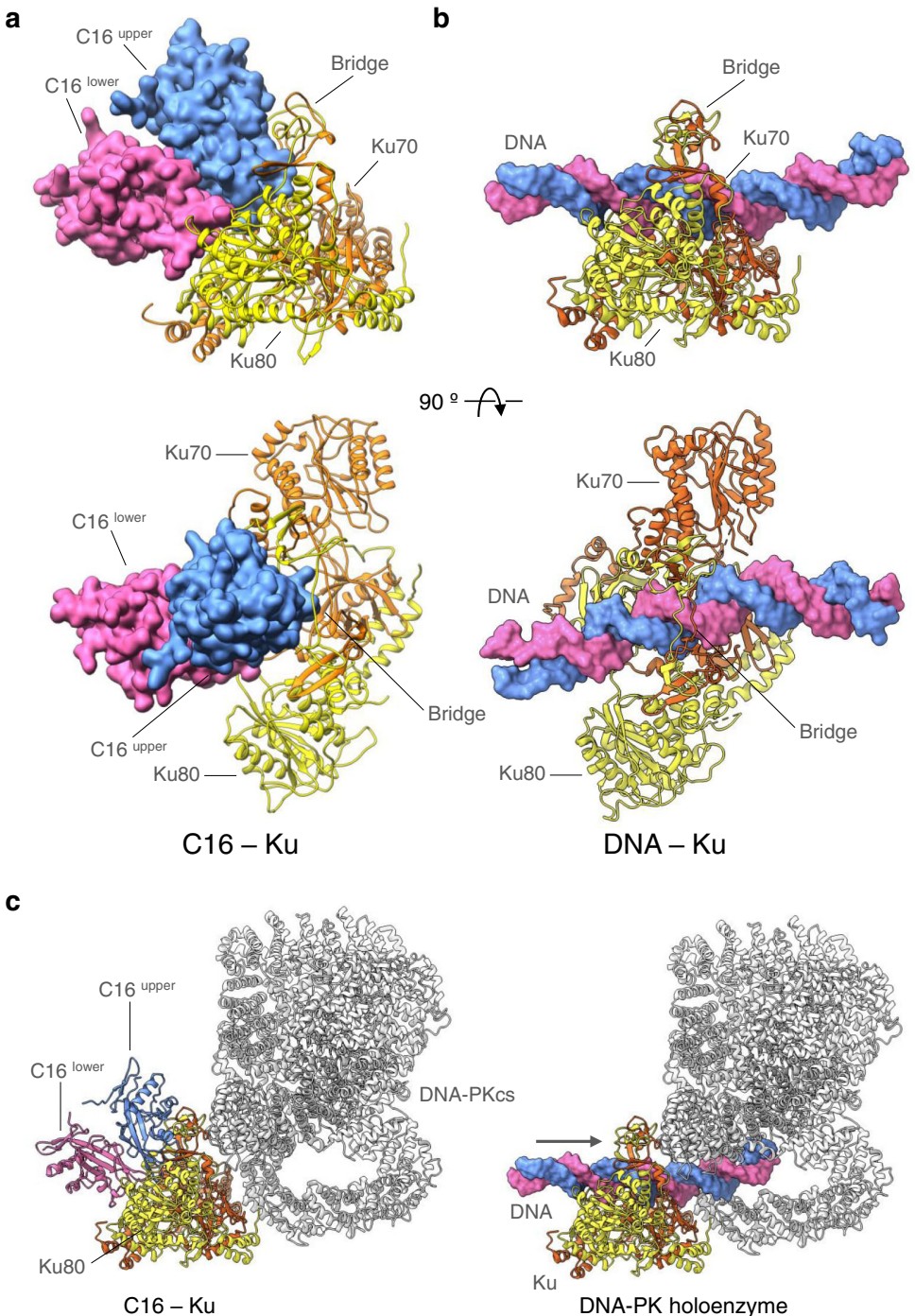

**Fig. 5 | C16 sterically blocks DNA binding. a** Structure of the C16-C–Ku complex obtained by cryoEM to be compared with b. Ku is shown in ribbons and C16-C is represented as space-filling model. **b** Ku bound to an extended DNA molecule in the context of its complex with DNA-PKcs (PDB code 5Y3R)(DNA-PKcs is not represented in the panel)[8] to be compared with a. The C16 dimer blocks the path of the DNA through the Ku bridge. Bottom panels correspond to the views rotated by 90° around the horizontal respect to the views in the top panels. **c** C16 modelled into the DNA-PK holoenzyme complex (left) would effectively block the insertion of a dsDNA end through Ku into the DNA-PK catalytic subunit (right), which activates DNA-PKcs activity as a protein kinase.

We analysed if AF-multimer[24] predicts the interaction between C16 and Ku observed by cryoEM. Neither the dimerization of C16 or the interaction with Ku was predicted. Only when using 1 copy of C16 as an input, provided some solutions that showed contact between C16 and Ku. However, none of these reproduced the structure determined experimentally, although they did succeed in predicting that C16-C contacts Ku (Supplementary Fig. 6c). Similar results were found when testing C4 (Supplementary Fig. 6c).

## C16 dimers sterically obstruct the entry of DNA to Ku

Comparison of the C16–Ku complex (Fig. 5a) with the structure of Ku bound to an extended DNA molecule as part of a complex with DNA-PKcs complex[8] (Fig. 5b) shows that C16 prevents DNA binding by Ku by sterically blocking the path of DNA running through the Ku hetero-dimer. C16 dimerization is critical for this mechanism because only the lower C16 subunit interacts strongly with Ku80, placing the upper C16 subunit in position to project its negatively charged plug into the

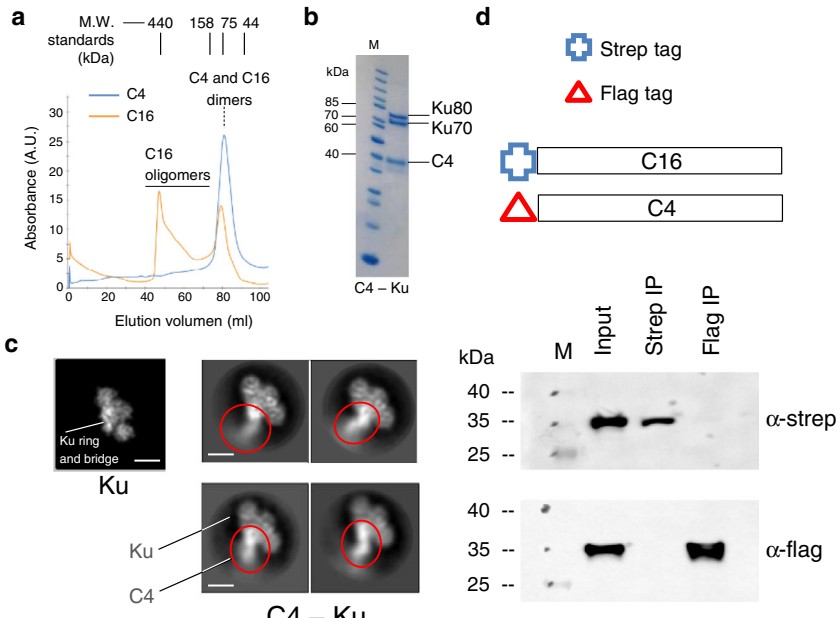

**Fig. 6 | Purification and cryoEM of the C4–Ku complex. a** SEC of C4 and C16 using a Superdex 200 16/600. To calibrate the column, we injected several proteins of known molecular weight, and their position is indicated: Ferritin (440 kDa) at 44 ml, Aldolase (158 kDa) at 73 ml, Conalbumin (75 kDa) at 80 ml and Albumin (44 kDa) at 91 ml. **b** The peak fraction of a SEC of the mixture of Ku and C4 was analysed by SDS-PAGE to confirm the ternary complex formation. Molecular markers (M) are shown in the left. The experiment shown in a and b was repeated twice with similar results. **c** Representative 2D averages of the cryoEM images obtained for C4–Ku (right panels) that can be compared with the computational projection of the structure of Ku (left panel). C4 attached to the region of the ring in Ku, as C16, but C4 appears to be very flexible in the complex. The projection of Ku was generated from the atomic structure of the Ku heterodimer lacking the SAP domain (PDB 1JEQ) but

filtered to 10 Å resolution. These projections lack the extra density observed in the images of C4–Ku which was interpreted as C4. Both galleries are shown at the same scale. Measuring bar corresponds to 50 Å. **d** Pull-down experiment showing that no heterodimers are detected when co-expressing C4 and C16. Proteins are tagged with a strep tag (Strep) or a flag tag (Flag). First lane is the whole cell lysate (input), second lane is the elution from strep-beads, third lane is the elution from flag-beads (IP stands for immunoprecipitation). Proteins were detected by western blot using anti-strep (α-strep) or anti-flag (α-flag) antibodies. Molecular weight (M) markers are indicated. Schematic representation of the tagging system employed in the C4 and C16 co-expression in *E. coli* is shown on the top panels. The experiment was repeated twice with similar results.

opening of the ring (Fig. 4d, e). When superimposed onto the full DNA-PK holoenzyme, it is clear that binding of the viral proteins would not directly compete with binding of Ku to the DNA-PK catalytic subunit, but would effectively block insertion of a dsDNA end into the holoenzyme, thereby preventing activation of DNA-PK signalling into the downstream innate immunity system (Fig. 5c).

**Vaccinia protein C4 recapitulates the mechanisms used by C16**

Another vaccinia protein, C4, also targets Ku and inhibits innate DNA sensing (Fig. 1). C4 and C16 share >53% identity in their C-terminal domains that interacts with the Ku heterodimer[11,12] (Supplementary Fig. 5b) and would therefore be expected to block DNA binding to Ku by a similar mechanism to C16. Purified C4 forms dimers, which elutes at a similar volume to C16 dimers in SEC, but does not assemble the larger oligomeric species observed with C16 (Fig. 6a). We purified the C4–Ku complex by simple mixing of the proteins and application to a SEC column as before. Here, C4 co-eluted with Ku and the peak fraction, containing the purified complex (Fig. 6b), was analysed by cryoEM (Fig. 6c).

2D averages of the C4–Ku complex, once compared with computational projections of the structure of Ku after filtering to the resolution of the cryoEM images, reveal that C4 attaches to Ku at the region of the ring. This is similar to C16 and consistent with the high conservation between the C-terminal domains of C4 and C16. However, the density for C4 in 2D class averages appeared to be highly flexible, preventing the determination of the structure for the C4–Ku complex. One possibility to explain this enhanced flexibility could arise from differences within the N-terminal domains of C4 and C16. Notably, C4 lacks the first β-strand, which in C16 makes important contacts

between the two molecules of the dimer (Supplementary Fig. 8a, b). The N-terminal domain of C4 also lacks the lysine residues that line the cleft where we find an unattached peptide in C16 (Supplementary Fig. 5a, 8a). Thus, it is likely that differences in the N-terminal domains could make the attachment between the two molecules in the C4 dimer more flexible than in C16. Supporting the idea that structural differences between the N-terminal domains of C4 and C16 reflect functional differences in the cell, the N-terminal domain of C16, but not that of C4, has been reported to interact with PHD2[19].

In addition, we tested if C4/C16 heterodimers could be formed, since C4 and C16 share 54% amino acid identity, have a similar fold and dimerise. For this, *E. coli* cells were co-transformed with plasmids expressing C4 and C16 each harbouring a different affinity tag (Fig. 6d). We did not detect co-elution of C4 and C16 in pull-downs using either tag, suggesting that C4 and C16 do not form a heterodimer. This result complements a previous report where HEK293T cells infected with VACV expressed a tagged version of C4 at endogenous levels that co-purified with Ku but not with C16[12].

**Conservation of C4/C16–Ku contacts in other orthopoxvirus**

We did not find close structural homologues of C16-C in the human genome or in the AlphaFold database of computational predictions, but the sequence of C4 and C16 are conserved in other viruses of the poxviridae family. We analysed the regions of interaction between C16 and Ku in the structure of the C16–Ku complex, and their conservation in these viruses (Supplementary Fig. 9). The acidic plug performs a different function in each subunit of the C16 dimer. The upper C16 subunit projects residues D245-L255 into the entry to the Ku ring whereas these residues bind Ku80 in the lower C16 subunit (Fig. 7a).

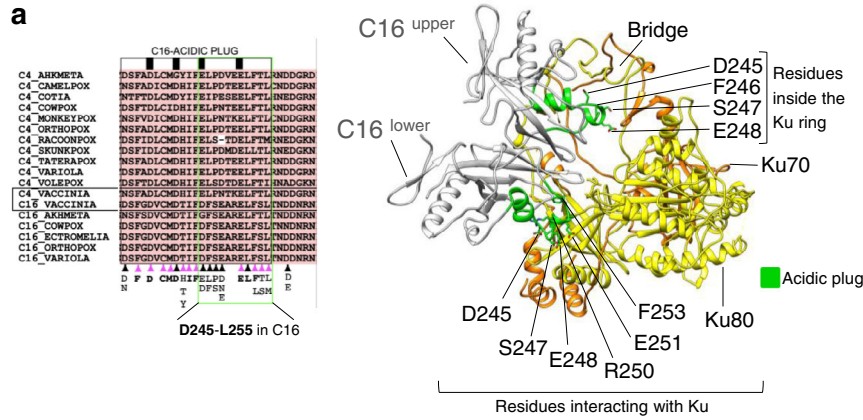

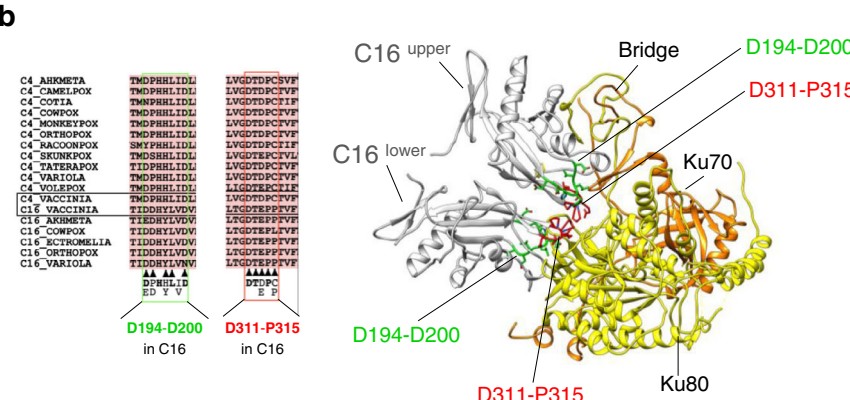

**Fig. 7 | Conservation of regions in C4/C16 involved in the interaction with Ku. a** Left panel: multiple amino acid sequence alignment of C4 and C16 proteins from the poxviridae family (Supplementary Fig. 9a for details). Amino acids involved in interactions with Ku are marked with filled triangles and those conserved in the acidic plug but not directly involved in the interaction with Ku are marked with a pink triangle. The predominant amino acid found at each site of interaction is also shown, with those shown in bold text indicating absolute/strong conservation of identity across both C4 and C16 proteins. The residue range corresponding to the described acidic plug of C16 is enclosed by a black rectangle, with regions of conserved negative charge indicated by blocks coloured black. Right panel: residues from the acidic plug are shown in the structure of C16–Ku. **b** Left panel: as in a but showing regions of C16 outside the acidic plug that interact with Ku, indicated within green and red boxes. Right panel: residues from these two regions are indicated in the structure of C16–Ku.

Most of these residues show strong conservation in C4 and C16 in several viruses, which extends to other amino acids forming the acidic plug (Fig. 7a, Supplementary Fig. 9). The structure of C16–Ku identifies two additional patches of residues (D194-D200 and D311-P315) that interact with Ku in both subunits of C16, and which are highly conserved across several of the viruses analysed (Fig. 7b, Supplementary Fig. 9).

## Discussion

The innate immune system responds to infection by recognizing molecular patterns that are unique to pathogens[1,2]. With viruses such as VACV, which replicate their dsDNA entirely in the cytoplasm of the host cell, detection of dsDNA within the cytoplasm becomes a critical component of the innate immune response[3,17]. The Ku heterodimer and the associated DNA-PK catalytic subunit, which are more usually associated with DNA repair functions within the nucleus[4–6], also play a key role in sensing viral DNA present in the cytoplasm. In response to this, VACV had evolved two closely related proteins, C4 and C16, that counteract DNA detection by Ku and DNA-PK[9–12]. Orthologues of C16 and/or C4 are identified in other orthopoxviruses such as variola (VARV)—the causative agent of smallpox, and monkeypox (MPXV). For example, MPXV C4 protein is 94.9 identical in sequence to VACV C4 (Supplementary Fig. 9). A number of other viruses also produce proteins that interact with components of the NHEJ pathway, including adenoviruses, Epstein-Barr virus, herpes simplex virus 1, human

papillomavirus and several others[26]. Although several different mechanisms seem to be in place, this strengthens the notion that interference with NHEJ proteins is a strategy used by many viruses to benefit their life cycle.

Here, we provide the structural basis for one such viral defence mechanism, in which the homologous C4 and C16 proteins share a C-terminal domain with more than 50% sequence identity that engages with the Ku heterodimer to prevent its binding to dsDNA and thus downstream signalling by DNA-PK, by sterically blocking the entry of DNA to Ku.

Binding of C16 involves a region of Ku80 conserved across both prokaryotic and eukaryotic Ku proteins, and that does not involve or require the N-terminal von Willebrand domain found only in the eukaryotic examples[27]. This therefore suggests that C16 and C4 may have arisen at a very early stage of cellular evolution, in which cytoplasmically replicating pox viruses such as vaccinia, may have infected non-nucleated precursors of the modern eukaryotes that form their current hosts[28]. Dimerization of C16 appears to be critical for its function, as only one of the subunits makes substantial interactions with Ku80 in order to place the other subunit in the correct position to project a highly negatively charged plug inside the ring opening of Ku, effectively blocking the access to DNA. C4 recapitulates much of this mechanism since its C-terminal domain is closely similar to C16, forming a dimer and interacting with Ku around the ring region.

Viral genomes are generally believed to be extremely parsimonious[29], so the presence of two homologous proteins that apparently fulfil the same function through strongly conserved domains and very similar interactions with their common target is perhaps surprising. Genetic studies indicate that C4 and C16 are only partially redundant in their ability to counteract the innate response to viral DNA[11,12]. The N-terminal domain of C16 but not C4 binds PHD2 and induces a hypoxic response during normoxia[19] and a reprogramming of central energy metabolism[30]. Therefore, the less strongly conserved N-terminal domains contribute to some distinct functions of C4 and C16 additional to their common ability to bind Ku and block dsDNA binding. The two proteins display differing biochemical behaviours, with C16 having the unique capacity to assemble larger-order multimeric species via self-association, even when bound to the Ku heterodimer. Whether and how this capacity for oligomerization contributes to antagonize the host innate immunity response, remains unknown. The structure of C16 reveals a positively charged cleft generated though dimerization of the N-terminal domain, which encloses a region of unassigned density in maps that suggest that this region may be involved in protein-protein interactions. Some credence is added to this idea, by the observed differences in N-terminal domains of the two viral proteins, which likely reflect their specific cellular functions[19,30].

Our work provides structural information that should aid in the design of small compounds that target the surface of interaction between C4/C16 and the Ku heterodimer and/or mimic the way C4 and C16 recognize Ku. Such molecules might also find utility in the treatment of human diseases associated with dysregulation of DNA sensing, for example, Aicardi-Goutieres syndrome. Ku and the DNA-PK holoenzyme function as DNA sensors for other viruses such as hepatitis B, so exploring how DNA is detected for these pathogens, and if other viruses have evolved proteins to evade the action of DNA-PK will be helpful to understanding the innate immune response.

## Methods

### Plasmids

Complete genome from vaccinia virus strain Western Reserve (WR) was a gift from Professor Antonio Alcamí (Spanish National Research Council, CSIC). First of all the Multiple Cloning Sites (MCS) from a standard pET-52b(+) plasmid (Novagen) was modified so it consisted of a TEV cleavage site followed by a Twin-Strep tag. This modification was done using a restriction-free cloning method[31] and a gBlocks™ gene fragment (IDT) containing the modified version of the MCS. C16 ORF was cloned in the modified pET-52b(+) plasmid using the restriction-free cloning method previously cited. The final construct [pET-52b(+)-FLC16-TEV-Twin-Strep] produced full-length C16 with a C-terminal Twin-Strep tag that could be removed upon incubation with the TEV protease. Supplementary Table 4 indicates the sequences of DNA fragments used in this work.

### Expression and purification of Ku, C4, C16 from baculovirus

C4 and C16 genes were synthesised by IDT as codon-optimized versions for expression in insect sf9 cells. Both C4 and C16 were cloned into a modified pFBDM vector harbouring a N-terminal Twin-Strep tag followed by a PreScission protease site and an uncleavable C-terminal 8xHistidine tag. These constructs were transposed into DH10Bac cells and the isolated bacmids were transfected into insect sf9 cells to generate the viral stocks. For protein expression, C4 or C16 viruses were used to infect, with a MOI of 2, High Five baculovirus cells grown in Insect-XPRESS protein-free medium (Lonza) to a density of 2 million. Expression was performed at 28 °C for 3 days and the cells were harvested by centrifugation at $1000 \times g$ during 15 min.

The C4 pellet was resuspended in lysis buffer [50 mM HEPES-KOH, pH 8.0, 500 mM NaCl, 10% (w/v) glycerol, 0.5 mM TCEP] supplemented with cOmplete™ EDTA-free protease inhibitors (Roche) and TURBO DNase (Life Technologies). Cells were lysed by Dounce homogenization and, after a 20-min incubation in ice, the slurry was clarified by centrifugation at $50,000 \times g$ for 60 min, passed through a 0.5 µm filter and loaded into a 5-ml Hitrap-Talon crude column (Cytiva) equilibrated with lysis buffer supplemented with 5 mM imidazole. A gradient elution in 10 column volumes up to 300 mM imidazole was performed and the fractions were analysed by SDS-PAGE in order to select and pool together those enriched in C4, which were subsequently loaded into a 5-ml Strep-Tactin XT column (IBA), previously equilibrated in 50 mM HEPES-KOH, pH 8.0, 250 mM NaCl, 5% (w/v) glycerol, 0.5 mM TCEP. C4 was then eluted in step with the same buffer supplemented with 50 mM biotin. Purity of full-length C4 was checked by SDS-PAGE and fractions of interest were pooled together and concentrated using an Amicon® Ultra-15 (Merck Life Sciences) filter with 30 kDa cut-off. Quality of the concentrated C4 protein was further assessed by size exclusion chromatography (SEC) in a Superose 6 Increase 3.2/300 column (Cytiva) equilibrated in buffer 50 mM HEPES-KOH pH 7.9, 250 mM NaCl, 1 mM EDTA, 0.5 mM TCEP and 0.5 mM PMSF. C16 from baculovirus was purified as described for C4. This protein was used in EMSA assays.

The Ku heterodimer was purified similarly to C4 and as described before[32]. Briefly, Ku heterodimers consisting of full-length Twin-Strep tagged Ku70 and full-length 10xHis tagged Ku80 were co-expressed using a baculovirus system. The tags were placed at the N-terminus in both proteins. Cell lysate was clarified by centrifugation and the Ku complex purified by immobilised metal-chelate chromatography. After protein elution using a buffer containing 300 mM imidazole, the fractions corresponding to the Ku complex were pooled and subsequently loaded into a 5-ml Strep-Tactin XT column (IBA). The eluted sample was next concentrated and loaded in a Superdex 200 (Cytiva) size exclusion chromatography. Peak fraction containing Ku70 and Ku80 were pooled, flash-frozen and stored for later use.

### Expression and purification of C16 for cryoEM

For C16 used in the cryoEM experiments, the protein was expressed and purified from *E. coli* using the pET-52b(+)-FLC16-TEV-Twin-Strep construct following a modified version of the protocol described in Peters et al.[11]. Expression was performed in *E. coli* Rossetta2(DE3) competent cells (Novagen) grown in standard Luria broth (LB) medium plus ampicillin and chloramphenicol. Protein expression was induced upon the addition of IPTG at 0.5 mM final concentration once cells had reached an optical density (600 nm) of 0.8–1.0 and temperature was stable at 16 °C. After overnight expression at 16 °C cells were pelleted and resuspended in buffer A (50 mM HEPES-KOH pH 7.5, 500 mM NaCl, 5% (w/v) glycerol, 0.5 mM TCEP), supplemented with lysozyme, benzonase® nuclease (Merck Life Sciences) and cOmplete™ EDTA-free protease inhibitor (Roche). Cells were lysed by sonication and cleared by centrifugation. The supernatant was loaded on a StrepTrap™ HP column (Cytiva) and bound protein was eluted in buffer A supplemented with 2.5 mM desthiobiotin. Fractions of interest were pooled together and incubated with His-TEV protease overnight at 4 °C. DTT was added to the mixture at 1 mM final concentration to improve efficiency of tag cleavage. The digested mixture was loaded on a HisTrap HP column (Cytiva) in order to remove His-TEV protease with the C16 protein eluting in the flow-through. The purity of the full-length C16 was checked by SDS-PAGE and fractions of interest were pooled together and concentrated using an Amicon® Ultra-15 (Merck Life Sciences) filter with 30 kDa cut-off. Quality of the concentrated C16 protein was further assessed by size exclusion chromatography (SEC) in a Superose 6 Increase 3.2/300 column (Cytiva) equilibrated in buffer B (50 mM HEPES-KOH pH 7.9, 250 mM NaCl, 1 mM EDTA, 0.5 mM TCEP and 0.5 mM PMSF).

### Purification of C4–Ku and C16–Ku complexes

The C16–Ku complex was assembled after incubating C16 and Ku in a 3:1 molar ratio, respectively, for 15 min at room temperature. The

mixture was then further purified by size exclusion chromatography column in a Superose 6 Increase 3.2/300 column (Cytiva) equilibrated in buffer B. Fractions were checked by SDS-PAGE and those corresponding to the peak of the C16–Ku complex were concentrated using an Amicon® Ultra-0.5 ml (Merck Life Sciences) filter with 30 kDa cut-off and vitrified. Source data are provided with this paper showing an uncropped version of the SDS-PAGE showing the size exclusion chromatography column for C16–Ku. The C4–Ku complex was assembled after incubating C4 and Ku in a 4:1 molar ratio, respectively for 30 min at room temperature. The mixture was then further purified by SEC in a Superose 6 Increase 3.2/300 column (Cytiva) equilibrated in buffer B. Fractions were checked by SDS-PAGE and those corresponding to the peak of the C4–Ku were vitrified. In both cases, we optimised the ratio for the incubation of C16/C4 and Ku by performing titrations, starting equimolar and different excess values. A molar ratio of 3:1 for C16:Ku and 4:1 for C4:Ku were the ones that achieved more complex and better separation from uncomplexed proteins in each case.

## Cryo-electron microscopy and image processing

Concentrated fractions of the C16–Ku complex obtained after SEC were applied to glow-discharged holey carbon grids (Quantifoil R 0.6/1 Cu 300 mesh) and vitrified using the Vitrobot Mark IV (Thermo Fisher Scientific) set to 4 °C and 100% nominal humidity. Vitrified grids were first analysed in the 200 kV JEM-2200FS (JEOL) equipped with a K3 direct electron detector (Gatan) available at the Electron Microscopy Unit from the CNIO (CNIO data set). Next selected grids were clipped and analysed in the 300 kV Titan Krios G3 electron microscope (ThermoFisher Scientific) using a K3 direct electron detector and operated in the Leicester Institute of Structural and Chemical Biology (UK, Leicester data set). Data acquisition parameters are summarized in Supplementary Table 1. Data collection for the Leicester data set was performed using the EPU software (ThermoFisher Scientific). Data collection for the CNIO data set was performed using the SerialEM software (https://bio3d.colorado.edu/SerialEM/).

Both data sets, CNIO and Leicester, were processed in a similar way using Relion[33,34] (Supplementary Table 2). Movies were corrected for drift (5 × 5 patches) and dose-weighted using Relion's own implementation of the MotionCor2 algorithm[35], and the contrast transfer function parameters were determined with CTFFIND4 (version 4.1)[36]. For the CNIO data set initial particles were firstly selected using the Laplacian-of-Gaussian blob detection algorithm available in Relion. Best classes were then used for Topaz training and picking[37] and ~1 million particles were extracted and extensively analysed in 2D. Of those, 308,714 particles were further analysed in 3D, first by generating ab initio volumes, and then by classification in 3D. Finally, 242,827 shiny particles were refined to generate a cryo-EM map with an estimated average resolution of 4 Å using gold-standard refinement methods and the Fourier shell correlation (FSC) cut-off of 0.143.

The trained topaz model from the CNIO data set was used for particle picking in the Leicester data set and ~12.5 million particles were extracted at 2.505 Å/pixel and thoroughly analysed in 2D resulting in 2.8 million particles that belonged to classes with clear secondary structural features. Those 2.8 million particles were subsequently analysed in 3D, generating ab initio starting volumes that were used for classification in 3D. A homogeneous particle subset was then extracted at the original pixel size (0.835 Å/pixel) and further classified in 3D. The resulting 845,006 particles were subsequently analysed in cryoSPARC[20], revealing the presence of a class that lacked C16 N-terminal domains. Those particles were removed from the initial 845,006 particles, and the resulting ~579,000 particles were further processed in Relion through Bayesian polishing, CTF and aberration refinement and generating a map (C16– Ku map) that exhibited an estimated average resolution of 2.46 Å (gold-standard FSC = 0.143). However, in this structure there were several regions that lacked detail due to intrinsic flexibility. These regions corresponded to the C16 N-terminal domains, to the binding interface between the C16 C-terminal domains and Ku, and to the DNA binding ring in Ku. In order to reveal detail in these three regions, we applied several strategies including the use of masks of the area of interest, local searches during 3D classification, the use of high T values during classification in 3D but without performing image alignment, and particle subtraction prior 3D classification. For the region corresponding to the C16 N-terminal domains an improved map named C16-N was generated with an estimated average resolution of 3.47 Å (gold-standard FSC = 0.143). An improved map at the binding interface between C16 C-terminal domains and Ku was also generated, named C16-C–Ku complex with 2.72 Å estimated average resolution (gold-standard FSC = 0.143). A composite cryoEM map was then built by merging C16-C–Ku and C16-N selecting one possible orientation between the flexible N- and C-terminal domains of C16. Finally, density in the ring region of Ku at the opposite end of C16 was also analysed, revealing the random orientation of a density that was connected to Ku70 and hypothesized as corresponding to the N-terminus of Ku70 that is not present in the available atomic structures.

2D classes corresponding to C16 were detected when performing 2D analysis of the initial 12.5 million particles from the Leicester data set corresponding to the C16–Ku complex. These images were assigned to C16 because they were identical to images obtained for purified C16 in a 200 kV JEM-2200FS (JEOL) available at the Electron Microscopy Unit from the CNIO. 319,596 particles were classified in classes corresponding to only C16, in which C16 forms stacks with a different number of layers and that were compatible with different oligomeric species. Sharpening for all the final maps was performed using DeepEMhancer software[38]. Local resolution ranges were analysed within Relion[33].

## 3D variability analysis

The resulting 579,000 particles from cryoSPARC were also analysed using 3D variability tools to characterize the amount of flexibility present in the C16–Ku complex[20,39]. Flexibility was decomposed in two eigenvectors filtered to 4 Å resolution, and the volume series for one of the components were visualized in different orientations and recorded in UCSF Chimera[40]. Supplementary movies 1, 2 and 3 reveal the flexibility present at the binding interface between C16 and Ku.

## Model building

The atomic structure of the Ku heterodimer bound to DNA 1JEY[25] and a prediction of the structure of C16 obtained using AlphaFold[21] were initially fitted into the cryoEM map of the C16-Ku complex using UCSF Chimera[40]. Ku70 and Ku80 can be unambiguously identified in cryoEM maps, due to the resolution attained and the known structural differences between the two proteins; the Ku heterodimer is not perfectly symmetrical and fitting of the Ku X-ray crystal structure 1JEY[25] into the cryoEM map to begin modelling provides only a single compatible orientation.

The fit to the experimental volume of all the docked polypeptide chains was adjusted manually in Coot[41] and the global fit optimised using phenix.refine[42]. Parameters defining the quality of the final atomic model are provided in Supplementary Table 3. The same procedure was applied to the C16 N-terminal domain. For the complete model comprising full-length C16 bound to the Ku heterodimer, a composite map was created using phenix.combine_focused_maps and the models for C16-N and C16-C–Ku to align it. Several rounds of manual adjustment in Coot and fit optimisation in phenix were performed to improve the quality of fit of the final complete atomic model (Supplementary Table 3).

## Testing the formation of C4/C16 heterodimers

BL21 (DE3) *E. coli* cells were co-transformed with both pET17-twin-strep-C16 and pET28b-flag-C4 and grown in standard LB medium

supplemented with ampicillin and kanamycin. Protein expression was induced upon the addition of IPTG at 0.5 mM final concentration once cells had reached an optical density (600 nm) of 0.8–1.0 and the temperature had been reduced from 37 to 16 °C. After growth over-night, cells were pelleted by centrifugation and then resuspended in buffer A (50 mM HEPES-KOH pH 7.5, 500 mM NaCl, 5% (w/v) gly-cerol, 0.5 mM TCEP), supplemented with lysozyme, benzonase nucle-ase (Merck Life Sciences) and cOmplete™ EDTA-free protease inhibitor (Roche). Cells were lysed by sonication and cleared by centrifugation.

Pull-downs were performed using the cell lysate from the C16 and C4 co-expression. The lysate was divided in half, and then applied to either Strep-Tactin beads (IBA, catalog number 2-1613-002, 1 ml to obtain a bed volume of 50 μl) or Anti-Flag beads (Merck, catalog number M8823, 100 μl to obtain a bed volume of 50 μl). After extensive washing, proteins retained on the beads were eluted with the addition of either biotin or flag peptide respectively and the eluted proteins analysed by SDS-PAGE and immunoblotting with either anti-strep (Merck, catalog number SAB27002216, 1:7000 dilution) or anti-flag antibodies (Merck, catalog number F3165, 1:5000 dilution).

### Sequence analysis of C4 and C16
Amino acids sequences corresponding to C4 and C16 proteins were identified by Protein BLAST search (https://blast.ncbi.nlm.nih.gov/Blast.cgi) and aligned using M-Coffee[43] (https://tcoffee.crg.eu/apps/tcoffee/do:mcoffee). Interactions between the C16 dimer and Ku were characterised through use of the online tool PDBsum (http://www.ebi.ac.uk/thornton-srv/databases/cgi-bin/pdbsum/GetPage.pl?pdbcode=index.html).

### Reporting summary
Further information on research design is available in the Nature Portfolio Reporting Summary linked to this article.

## Data availability
The cryoEM maps for the C16 – Ku complex generated in this study have been deposited in the Electron Microscopy Data Base (EMDB) (https://www.ebi.ac.uk/emdb/) under accession codes EMD-15414, EMD-15415 and EMD-15416. The atomic models for the C16 – Ku complex generated in this study have been deposited in the Protein Data Base (PDB) (https://www.rcsb.org/) under accession codes 8AG3, 8AG4 and 8AG5. Source data are provided with this paper.

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

## Acknowledgements

We acknowledge the help of Dr Christos Savva from the Leicester Institute of Structural and Chemical Biology (UK) for assistance with data collection, and the help of Clara Santiveri and Ramón Campos from the Spectroscopy and Nuclear Magnetic Resonance Unit at CNIO for their assistance in the Mass Photometry experiments. We thank Tomás de Garay and REFEYN (Oxford, UK) for their assistance during mass photometry measurements. We acknowledge the support of the National Institute of Health Carlos III to CNIO. This work was funded by the Autonomous Region of Madrid and co-funded by the European Social Fund and the European Regional Development Fund [Y2018/BIO-4747 and P2018/NMT-4443 to O.L], and by Cancer Research UK Programme Grants C302/A14532 and C302/A24386, to A.W.O and L.H.P.

## Author contributions

A.R.-C. performed all experiments related with C16–Ku and cryoEM and contributed to the purification and cryoEM of C4– Ku; R.A.-B. performed purification and cryoEM of C4–Ku and the EMSA experiments; A.R.-R. and P.E.-B. helped in the production of C16; J.B. and R.F.-L. contributed to data collection using the JEM-2200FS (JEOL) microscope at CNIO; L.H.P., A.W.O. and O.L. designed the research; L.H.P. and O.L. prepared the first versions of the manuscript.

## Competing interests

The authors declare no competing interests.
