## [Peer Review File · Nature Communications]

Structural basis for the inactivation of cytosolic DNA sensing by the vaccinia virusREVIEWER COMMENTS

Reviewer #1 (Remarks to the Author):

Review

In the article "Structural basis for the inactivation of cytosolic DNA sensing by the vaccinia virus", A. Rivera-Calzada et al present the cryo-EM structure of Ku – C16 complex at high resolution, some biochemical analyses of the Ku-C16 and Ku-C4 complexes and some preliminary data on the Ku-C4 complex. They nicely show that C16 binds as a dimer with the C-terminal region of one subunit interacting with the Ku80 face of the heterodimer and a second subunit that is positioned in the middle of Ku ring and thus prevents DNA binding. This result is in agreement with EMSA experiments that show an inhibition of Ku-DNA interaction with C16 at nM concentrations. The results are important both for the virology and DNA repair field. They are timely since they provide clear molecular basis for the first time for a central function of Ku in the cytosol as DNA sensor of vaccinia virus replicating its genome. The article is well written and the figures are of high quality. The manuscript should be reinforced more in-depth analyses in particular of AlphaFold results as well as multiple sequence alignments of C16 and C4 orthologs. These points are detailed below as well as other points that should be clarified.

Major points:

- 1) The Cryo-EM suggests that the main complex observed is one Ku and two C16 proteins. The authors should further document this point with biochemical or biophysical approaches. They should present mass photometry data (already used for C16) with Ku-C16 and Ku-C4 complexes in presence of several C16 and C4 concentrations.
- 2) Page 5 'a prediction from AlphaFold 15 was used as starting point for modeling the atomic structure of the C16'. The authors may present data to evaluate the quality of the AlphaFold (AF) model. They may comment if at the end of the cryo-EM refinement and building, there were differences between the cryo-EM model and the AF models of C16 used initially.
- 3) The authors should present the results of AlphaFold-Multimer with the different complexes discussed in the manuscript. It is important to know if AF succeeds in prediction Ku-1C16 or Ku-2C16 complexes and if yes to compare with the Cryo-EM structure. They should try also prediction the Ku:1C4 or Ku:2C4 complexes. Finally, they should test AF with the 2C16 and 4C16 complexes. This is important from a methodological point of view but also to see if it reinforces some hypotheses like acidic "plug".
- 4) Page 6 the authors may analyze by mass-spectrometry the C16 dimer to identify the nature of the ligand observed by cryo-EM.
- 5) Page 6 – ext Data fig 5c : Have the authors tried to determine the cryo-EM structure of the tetramer of C16. They present nice 2D class in the middle of Ext Fig 5c.
- 6) Page 7 "certain swinging movement of C16, which can result in some deformation of the bridge forming the DNA binding ring". From the movies, the reader will have an approximate idea of C16 swinging and the deformation of the Ku bridge is difficult to evaluate. The authors should add figures of superimposition illustrating how does the bridge of Ku change of conformation upon C16 binding compare to the apo-form (1JEQ). They may also illustrate with figures C16 conformational changes to complement the movies.
- 7) Page 7 " it projects an intensely negatively charged 'plug' formed by a helix-turn-helix motif from residues 233-254 into the mouth of the bridge, which itself displays a positively charged surface (Fig. 4f)." The authors pass super rapidly on this point that is quite new and original in Ku-protein

interactions. They may comment if this region is conserved in other C16 orthologs. They may describe the interface a little more.

8) Page 8 "Comparison of the C16 – Ku complex (Fig. 5a) with the structure of Ku bound to an extended DNA molecule 8 (Fig. 5b)" The authors should precise that they compare with a structure of Ku-DNA-PKcs bound to an extended DNA molecule and in the legend of the Figure that they removed DNA-PKcs on the Ku70 side. The reader can understand that a structure was determined with Ku in complex with a large DNA.

9) How do the author attribute the Ku70 and Ku80 subunits in the cryo-EM maps? Ku is pseudo-symmetrical molecule. They should describe how they succeed to attribute experimentally the Ku70 and Ku80 subunits in the cryo-EM maps.

10) The authors should reinforce the sequence alignments information. In Ext Data, they present alignments with 3 sequences of C4 and 2 of C16. They should present sequence alignments with more orthologs sequences from other virus. In particular, it is important to know in which virus the two main patches of interactions with Ku are conserved or and in which these patches are not conserved.

11) Mechanism of C16. From the EMSA, the authors clearly show that when C16 is complexed with Ku, it prevents DNA binding in their assay. This suggests a mechanism where C16 will fix Ku molecules and then prevents DNA sensing. What happens when Ku is first on DNA? Can C16 outcompete with DNA for Ku? We suggest to test by EMSA, if pre-formed Ku-DNA complex can be weakened by the presence of C16.

Minor points:

1) The authors may explain why they assembled C16 and Ku with a 3:1 molar ratio and C4 and Ku in a 4:1 molar ratio. How do they arrive to this protocol or the rationale behind?

2) Fig 3 legend : repetition of the same sentence

3) SEC: Authors may add on Fig2a, Fig 6a and Ext Data Fig1a arrows above the SEC chromatograms indicating the elution time of proteins used for calibration?

4) Ext Dat Fig 1b and text: the authors may precise in Mat&Method or legends what they mean by non-stringent denaturing conditions.

5) The authors may precise the theoretical MW of the Ku-C16 complexes beside fig 2a for example (they mentioned bottom of page 4 "represent a roughly 300 kDa complex", to which stoichiometry could it correspond)

Reviewer #2 (Remarks to the Author):

This nice paper reports the cryoEM structure of the vaccinia virus (VACV) protein C16 in complex with the Ku heterodimer Ku80 and Ku70. A structure of the related VACV protein C4 in complex with the Ku complex is found to be similar. The structural data are novel and not only provide the structure of the VACV C16 / C4 proteins, but also reveal the mechanism by which C16 / C4 prevents Ku binding DNA and so functioning in DNA sensing via DNA-PK. This is generally a well written paper.

Issues for the authors to address:

1. The authors should provide a fuller introduction / abstract that sets the present findings in context and refers better to existing literature. For examples:

- a. The introduction and abstract mention only VACV proteins C16 and C4 as proteins that “inactivate DNA sensing”, and do not mention other VACV strategies to do this. For instance, the VACV B2 protein, called a poxin / schlafen, is a phosphodiesterase and cleaves cGAMP. The primary papers should be cited. Also relevant, is the fact that VACV encodes many other inhibitors of IRF3 or NF-kB signalling that inhibit the pathways activated by DNA-PK. Reference to this by a review article might be appropriate.
 - b. The primary literature reporting DNA-PK as a DNA sensor should be cited. Zhang et al., 2011 and Ferguson et al., 2012. Also this reviewer feels that the word “especially” before DNA-PK on line 6 should be deleted because it gives an unjustified emphasis.
 - c. Intro, para 2, line 4, Also cite ref 25. (Ref 9 is more about C16 inducing a hypoxic response). Ditto, next para line 3.
 - d. Para 3, lines 8-9. The literature characterising the proteins C4 and C16 and reporting their contribution to virulence from VACV strain WR should be cited. Fahy et al JGV, 2008; Ember et al., JGV 2012; ref 10.
 - e. The introduction needs to state what is already known about C16 and C4. Specifically, ref 25 showed that: 1) C16 binds Ku directly, 2) C16 did not bind to DNA-PKcs, 3) C16 prevented Ku binding to DNA, 4) C16 prevented Ku interacting with DNA-PKcs, 5) C16 did not bind DNA itself, and 6) the Ku binding activity of C16 resides in the C-terminal domain. For C4, ref 10 showed that 1) C4 binds Ku, 2) C4 does not bind DNA-PKcs, 3) C4 does not bind DNA, 4) C4 prevents Ku binding to DNA, and 5) the C-terminal region of C4 binds Ku and 6) is sufficient to inhibit DNA sensing. When all this is stated upfront, it is evident that the data shown in Fig. 1, are in most part redundant.
2. Nomenclature: confusingly, the protein named C16 differs in the different VACV strains, so it is necessary to make explicit that the C16 protein studied here is from VACV strain Western Reserve. Monkeypox and poxviruses should each be one word, not two. Epstein-Barr virus (add hyphen).
 3. Ref 25 reported that the MVA version of C16 has an internal deletion of amino acids 277-281, and this very largely prevents its binding with Ku. Could the authors comment on this from the structural perspective?
 4. Page 5 line 14: the Ku resolution was 2.5A, please also state what it was for the C16 C-terminal domain?
 5. Does the C16 C-terminal domain dimerise in the absence of the N-terminal domain? One would surmise yes, since dimerisation is necessary for function, and expression of the C-terminal domain of either C4 or C16 alone can bind Ku. But was this tested directly?
 6. Since C4 and C16 share 54% aa identity, have the same fold, both dimerise and bind the same target protein in a conserved manner, can heterodimers be formed?
 7. Is the C16/C4 fold detected by alpha fold in other proteins from viruses that interact with the NHEJ pathway or are sensed by DNA-PK?
 8. Page 10, para 3. The sentence starting line 3 and finishing “ ... suggesting that each protein might perform some distinct function additional to their common ability to block dsDNA binding to Ku, perhaps mediated by their less strongly conserved N-terminal domains” needs modification. It is an established fact that the N-terminal domain of C16 binds PHD2 and induces a hypoxic response during normoxia (ref 9) and induce a reprogramming of central energy metabolism (Mazzon et al., JGV 2015), whereas C4 does not do this. These studies should be mentioned in this context.
 9. Fig 6c. The measuring bar referred to is absent.

Reviewer #3 (Remarks to the Author):

Rivera-Calzada et al. report the inhibitory effects of vaccinia virus proteins C16 and C4 on DNA-end sensing by host Ku protein in the cytoplasm and thus their ability to block the innate immune response. The authors demonstrate the virus-encoded C16 or C4 protein each alone blocks DNA binding by heterodimeric Ku70/80 (Ku) and have determined cryoEM structures of the C16-Ku complex at near atomic resolution (2.5 – 3.5 Å). The structure reveals the physical interactions between C16 with Ku and how C16 blocks the DNA-binding channel of Ku. Even though a definitive structure of C4-Ku complex cannot be obtained, 2D projections of the C4-Ku complex indicate a similar mechanism of the viral protein and Ku association. The study is well executed, and the results interesting. After addressing a few minor points listed below, the manuscript will be suitable for publication in Nature Communication.

1. The first report that the C-terminal half of C16 blocks DNA binding by Ku and its effect on attenuating the host innate immune response was published by Peters et al. in 2013 (<https://pubmed.ncbi.nlm.nih.gov/24098118/>). Although the authors cited this paper in the Methods section for preparing C16 protein, Peters' paper should be cited at the very beginning and credited for their critical finding.
2. Could the authors please show the boundary between the N- and C-terminal domains of C16? Can the two domains of C16 be safely separated? Is the N-terminal domain of C16 necessary for inhibiting the DNA-binding activity of Ku? Peters et al. showed that the C-terminal domain. (aa 215 – 331) is sufficient and necessary for blocking Ku from DNA binding. But in the manuscript, the authors cited reference #10 and state that dividing C16 between residues 214 and 215 would fragment the C-terminal domain (p. 8).
3. Is there any evidence that DNA-PKcs, which forms a protein complex with Ku in nucleus is involved in cytoplasmic DNA sensing and innate immunity?
4. On p. 3, the authors refer to each subunit in the C16 dimer as "monomers". Shouldn't they be subunits of the dimeric C16?
5. When referring to the C-terminal half of C16 and C4, it may be clearer to distinguish them from the full-length protein by labeling them as C16-C and C4-C rather than C16 and C4.
6. C16 and Ku both are shown in cartoon or space-filling diagrams in all figures. It would help to show one in cartoon and the other in space-filling, for example, in Fig. 5 when comparing Ku binding to DNA or C16.

Point-by-point response to the reviewers

Reviewer #1

In the article “Structural basis for the inactivation of cytosolic DNA sensing by the vaccinia virus , A. Rivera-Calzada et al present the cryo-EM structure of Ku – C16 complex at high resolution, some biochemical analyses of the Ku-C16 and Ku-C4 complexes and some preliminary data on the Ku-C4 complex. They nicely show that C16 binds as a dimer with the C-terminal region of one subunit interacting with the Ku80 face of the heterodimer and a second subunit that is positioned in the middle of Ku ring and thus prevents DNA binding. This result is in agreement with EMSA experiments that show an inhibition of Ku-DNA interaction with C16 at nM concentrations. The results are important both for the virology and DNA repair field. They are timely since they provide clear molecular basis for the first time for a central function of Ku in the cytosol as DNA sensor of vaccinia virus replicating its genome. The article is well written and the figures are of high quality. The manuscript should be reinforced more in-depth analyses in particular of AlphaFold results as well as multiple sequence alignments of C16 and C4 orthologs. These points are detailed below as well as other points that should be clarified.

Thanks for the positive comments.

We have addressed all the points raised, which helped to improve the manuscript substantially. We have payed special attention to the questions regarding AlphaFold and the sequence alignments.

Major points:

1) The Cryo-EM suggests that the main complex observed is one Ku and two C16 proteins. The authors should further document this point with biochemical or biophysical approaches. They should present mass photometry data (already used for C16) with Ku-C16 and Ku-C4 complexes in presence of several C16 and C4 concentrations.

It is important to mention that the cryoEM map of C16 – Ku was determined at 2.5 Angstrom (average resolution), and at this resolution, the presence of two copies of the C-terminal domain of C16 forming a dimer interacting with Ku is defined without any doubt.

This is now indicated more explicitly in the revised version.

Therefore, the structure of the C16 – Ku complex demonstrates that it is a dimer of C16 that is in complex with Ku70/Ku80 without the minor ambiguity. With this in mind, determining the stoichiometry of the complex by biochemical and biophysical methods would complement the analysis, but in our view, it is not essential to conclude that the complex contains 1 Ku and 2 C16 proteins.

However, we have performed the suggested experiments. When C16 was incubated with Ku, the peaks corresponding to C16 reduced the number of counts, whereas a peak around 260 kDa increased the number of counts, which could correspond to the formation of the C16 - Ku complex.

We were not able to interpret these results unambiguously because C16 on its own (without Ku) generated a multitude of peaks due to its

oligomerization. When we performed the experiment of C4 and Ku, we also faced technical problems, and unfortunately, we have been unable to reach a conclusion.

Although, as mentioned above, we believe these experiments would complement the work but they are not critical to define the stoichiometry of the complex, and given our problems we encountered when using mass photometry, we have performed a size-exclusion chromatography experiment of a construct containing only the C16 C-terminal domain (**Supplementary Figure 5e**). This experiment shows that the C16 C-terminal domain forms dimers on its own, and it reinforces that C16 and C4 interact with Ku as a dimer since this is the domain that binds Ku and the most highly conserved between C4 and C16.

2) Page 5 'a prediction from AlphaFold 15 was used as starting point for modeling the atomic structure of the C16' The authors may present data to evaluate the quality of the AlphaFold (AF) model. They may comment if at the end of the cryo-EM refinement and building, there were differences between the cryo-EM model and the AF models of C16 used initially.

AF and AF-multimer predict the structure of the N- and C-terminal domains accurately, but predictions fail to predict the dimer interface that is found in the cryoEM structure.

This information is now included in the revised version of the manuscript as part of **Supplementary Figure 6a and 6b**.

3) The authors should present the results of AlphaFold-Multimer with the different complexes discussed in the manuscript. It is important to know if AF succeeds in prediction Ku-1C16 or Ku-2C16 complexes and if yes to compare with the Cryo-EM structure. They should try also prediction the Ku:1C4 or Ku:2C4 complexes. Finally, they should test AF with the 2C16 and 4C16 complexes. This is important from a methodological point of view but also to see if it reinforces some hypotheses like acidic "plug".

We have calculated all the predictions suggested by the reviewer.

AF-multimer was unable to predict the dimerization of C4 and C16. AF-multimer was unable to predict the interaction between C4 or C16 and Ku when using 2 copies of C16 or C4, and all solutions show C4 and C16 that are not in contact with Ku. Only when using 1 copy of C4 and C16, some solutions predict contacts between the C-terminal domain of C4 and C16 with Ku, but they contact in regions that are different to those determined in the experimental structures.

In conclusion, AF predicts well the individual domains of C16, but it fails to predict their dimerization and their interaction with Ku.

We have added this information in the text and as **Supplementary Fig. 6c**

4) Page 6 the authors may analyze by mass-spectrometry the C16 dimer to identify the nature of the ligand observed by cryo-EM.

The density seems to correspond to an alpha helix and we speculate that this could come from a neighboring C16 that it's multimerizing. So, mass spectrometry will not work if this is the case.

But we cannot exclude this is a ligand unrelated to C16.

Determining this might need substantial work that is out of the scope of this manuscript.

In the revised manuscript we indicate that the assignment of this density to C16 is an educated guess but purely speculative.

5) Page 6 – ext Data fig 5c : Have the authors tried to determine the cryo-EM structure of the tetramer of C16. They present nice 2D class in the middle of Ext Fig 5c.

Yes, we invested quite some time trying to determine the structure of these oligomers, unsuccessfully.

We collected data using a K3 direct electron detector and 2D averages were of good quality, but we were unable to obtain a reconstruction even at a moderate resolution.

Based on the problems we faced during image processing, we speculate that there could be a lack of sufficient views to determine the structure, that there is a mix of more than one form of oligomerization between C16 molecules, and/or there is conformational heterogeneity/flexibility.

Actually, the 2D averages (above) suggest some flexibility in the relative orientation between the two sides of the molecule.

We now indicate this in the revised version of the manuscript.

6) Page 7 “certain swinging movement of C16, which can result in some deformation of the bridge forming the DNA binding ring”. From the movies, the reader will have an approximate idea of C16 swinging and the deformation of the Ku bridge is difficult to evaluate. The authors should add figures of superimposition illustrating how does the bridge of Ku change of conformation upon C16 binding compare to the apo-form (1JEQ). They may also illustrate with figures C16 conformational changes to complement the movies.

We now show this in Supplementary Figure 7a.

7) Page 7 “ it projects an intensely negatively charged ‘plug’ formed by a helix-turn-helix motif from residues 233-254 into the mouth of the bridge, which itself displays a positively charged surface (Fig. 4f).” The authors pass super rapidly on this point that is quite new and original in Ku-protein interactions. They may comment if this region is conserved in other C16 orthologs. They may describe the interface a little more.

We now describe the interface with more detail in the text, and we have made new panels in the Supplementary Fig. 7b.

We also include a new section in Results for all the issues raised by the reviewer about the conservation of this and another features, which are part of a new Supplementary Fig. 9 and a new Figure 7.

8) Page 8 “Comparison of the C16 – Ku complex (Fig. 5a) with the structure of Ku bound to an extended DNA molecule 8 (Fig. 5b)” The authors should precise that they compare with a structure of Ku-DNA-PKcs bound to an extended DNA molecule and in the legend of the Figure that they removed DNA-PKcs on the Ku70 side. The reader can understand that a structure was determined with Ku in complex with a large DNA.

The reviewer is right and this is now fixed.

9) How do the author attribute the Ku70 and Ku80 subunits in the cryo-EM maps? Ku is pseudo- symmetrical molecule. They should describe how they succeed to attribute experimentally the Ku70 and Ku80 subunits in the cryo-EM maps.

The reviewer is correct that the Ku heterodimer is a pseudo-symmetrical structure. However, it is not perfectly symmetrical, and the unbiased fitting of the Ku crystal structure (PDB 1jeq; Walker et al. Nature 2001) into the cryoEM map to start atomic modelling identifies only one orientation compatible between the crystal structure and the cryoEM map.

What is more relevant for the question raised, the resolution of our cryoEM map (2.5 Angstrom) is similar to that of the crystal structure (Walker et al. Nature 2001). At this resolution there was no ambiguity to identify the differences between Ku70 and Ku80.

We clarify this with a sentence in the Methods section.

10) The authors should reinforce the sequence alignments information. In Ext Data, they present alignments with 3 sequences of C4 and 2 of C16. They should present sequence alignments with more orthologs sequences from other virus. In particular, it is important to know in which virus the two main patches of interactions with Ku are conserved or and in which these patches are not conserved.

We have performed the analysis suggested, and we have also prepared some panels showing the location of conserved residues in the structure.

This will help readers to appreciate the location of regions involved in the interaction between C16 and Ku and the residues that are part of the acidic plug in parallel to their conservation.

This is now part of a new section in Results and we have prepared two new figures, Supplementary Fig 9 and Figure 7.

11) Mechanism of C16. From the EMSA, the authors clearly show that when C16 is complexed with Ku, it prevents DNA binding in their assay. This suggests a mechanism where C16 will fix Ku molecules and then prevents DNA sensing. What happens when Ku is first on DNA? Can C16 outcompete with DNA for Ku? We suggest to test by EMSA, if pre-formed Ku-DNA complex can be weakened by the presence of C16.

This is an interesting suggestion. We have performed the experiment, which is now in Figure 1d in the revised version of the manuscript.

C4 and C16 are able to out-compete DNA binding when Ku was bound to DNA before incubation with C4 and C16.

Minor points:

1) *The authors may explain why they assembled C16 and Ku with a 3:1 molar ratio and C4 and Ku in a 4:1 molar ratio. How do they arrive to this protocol or the rationale behind?*

We did titrations for both cases, starting equimolar and different excess values. Those were the ones that achieved more complex and better separation from uncomplexed proteins in each case.

This is now indicated in Methods.

2) *Fig 3 legend : repetition of the same sentence*

Error fixed.

3) *SEC: Authors may add on Fig2a, Fig 6a and Ext Data Fig1a arrows above the SEC chromatograms indicating the elution time of proteins used for calibration?*

Arrows for the calibrations are now included.

4) *Ext Dat Fig 1b and text: the authors may precise in Mat&Method or legends what they mean by non-stringent denaturing conditions.*

In this SDS-PAGE, samples were mixed with “gel loading buffer” but the mix was not boiled before loading in the gel.

This is now indicated.

5) *The authors may precise the theoretical MW of the Ku-C16 complexes beside fig 2a for example (they mentioned bottom of page 4 “represent a roughly 300 kDa complex”, to which stoichiometry could it correspond)*

The theoretical weight of Ku – C16 is around 240 kDa (Ku70: 75.1 kDa; Ku80: 85.4 kDa; C16: 38.5 kDa per subunit).

We have rephased this part of Results since it is not correct that we estimated the molecular weight of Ku-C16 by its mobility in the SEC.

Reviewer #2

This nice paper reports the cryoEM structure of the vaccinia virus (VACV) protein C16 in complex with the Ku heterodimer Ku80 and Ku70. A structure of the related VACV protein C4 in complex with the Ku complex is found to be similar. The structural data are novel and not only provide the structure of the VACV C16 / C4 proteins, but also reveal the mechanism by which C16 / C4 prevents Ku binding DNA and so functioning in DNA sensing via DNA-PK. This is generally a well written paper.

Thanks for the positive comments.

We have addressed all the points raised, which helped to improve the manuscript.

Issues for the authors to address:

1. *The authors should provide a fuller introduction / abstract that sets the present findings in context and refers better to existing literature.*

We have improved the Introduction following the suggestions from the reviewer.

For examples:

a. The introduction and abstract mention only VACV proteins C16 and C4 as proteins that “inactivate DNA sensing”, and do not mention other VACV strategies to do this. For instance, the VACV B2 protein, called a poxin / schlafen, is a phosphodiesterase and cleaves cGAMP. The primary papers should be cited. Also relevant, is the fact that VACV encodes many other inhibitors of IRF3 or NF- κ B signalling that inhibit the pathways activated by DNA-PK. Reference to this by a review article might be appropriate.

Done

b. The primary literature reporting DNA-PK as a DNA sensor should be cited. Zhang et al., 2011 and Ferguson et al., 2012.

The reviewer is right. Done.

Also this reviewer feels that the word “especially” before DNA-PK on line 6 should be deleted because it gives an unjustified emphasis.

Corrected.

c. Intro, para 2, line 4, Also cite ref 25. (Ref 9 is more about C16 inducing a hypoxic response). Ditto, next para line 3.

Corrected.

d. Para 3, lines 8-9. The literature characterising the proteins C4 and C16 and reporting their contribution to virulence from VACV strain WR should be cited. Fahy et al JGV, 2008; Ember et al., JGV 2012; ref 10.

Done

e. The introduction needs to state what is already known about C16 and C4. Specifically, ref 25 showed that: 1) C16 binds Ku directly,

Done

2) C16 did not bind to DNA-PKcs,

Done

3) C16 prevented Ku binding to DNA,

Done

4) C16 prevented Ku interacting with DNA-PKcs,

Done

5) C16 did not bind DNA itself,

Done

and 6) the Ku binding activity of C16 resides in the C-terminal domain

Done

. For C4, ref 10 showed that 1) C4 binds Ku, 2) C4 does not bind DNA-PKcs, 3) C4 does not bind DNA, 4) C4 prevents Ku binding to DNA, and 5) the C-terminal region of C4 binds Ku and 6) is sufficient to inhibit DNA sensing.

Done

When all this is stated upfront, it is evident that the data shown in Fig. 1, are in most part redundant.

We have decided to keep the revised version of Figure 1. In addition to confirming the published results using the proteins we produce and use for the structural analysis, it adds that we observed no synergistic effect when C4 and C16 were present together, suggesting that they act through a similar mechanism in blocking the interaction of Ku with DNA.

Also, after a suggestion from reviewer #1, the figure is completed with a new experiment showing that C4 and C16 can out-compete DNA when the Ku-DNA complex is pre-assembled before adding C4 or C16.

2. Nomenclature: confusingly, the protein named C16 differs in the different VACV strains, so it is necessary to make explicit that the C16 protein studied here is from VACV strain Western Reserve.

The reviewer is correct that this is very important to clarify.

This is now indicated in Introduction.

Monkeypox and poxviruses should each be one word, not two. Epstein-Barr virus (add hyphen).

These errors have been fixed.

3. Ref 25 reported that the MVA version of C16 has an internal deletion of amino acids 277-281, and this very largely prevents its binding with Ku. Could the authors comment on this from the structural perspective?

Our structural data indicate that these residues (green colour in this figure) are not in contact with Ku.

A Cys-Tyr-Cys motif – residues 187-189 in C16 (red colour in this figure) - that also impacted in C16 function in the same manuscript, are also buried in the core of the C16 C-terminal domain. Therefore, the effect of these mutation on the interaction between C16 and Ku cannot be a direct impact on the interface with Ku, but some indirect effect, maybe by some destabilisation of the C16 C-terminal domain fold.

This is now commented in the Results section.

4. Page 5 line 14: the Ku resolution was 2.5Å, please also state what it was for the C16 C-terminal domain?

This information was presented in Supplementary Fig. 4a as a map that is colour-coded based on the resolution of each region.

To help readers, we now improve this figure by indicating the position of Ku and C16 C and N-terminal domains in the map.

As suggested, we also add the information about the resolution of C16 in the text.

5. Does the C16 C-terminal domain dimerise in the absence of the N-terminal domain?

One would surmise yes, since dimerisation is necessary for function, and expression of the C-terminal domain of either C4 or C16 alone can bind Ku. But was this tested directly?

C16 C-terminal domain dimerised in the absence of the N-terminal domain.

We have performed a size-exclusion chromatography experiment of a construct containing only the C16 C-terminal domain (**Supplementary Figure 5e**). This experiment shows that the C16 C-terminal domain forms dimers on its own, and it reinforces that C16 and C4 interact with Ku as a dimer since this is the domain that binds Ku and the most highly conserved between C4 and C16.

6. Since C4 and C16 share 54% aa identity, have the same fold, both dimerise and bind the same target protein in a conserved manner, can heterodimers be formed?

We have now tested this by co-expressing both proteins with a different tag, and the results are shown in the Figure 6d of the revised manuscript. Our results suggest that C4 and C16 cannot form heterodimers.

This agrees with previous reports where HEK293T cells infected with VACV that expressed a tagged version of C4 at endogenous levels which co-purified with Ku but not with C16 (Scutts et al, Cell reports 2018).

All this is now part of the Results section in the revised version of the manuscript.

7. Is the C16/C4 fold detected by alpha fold in other proteins from viruses that interact with the NHEJ pathway or are sensed by DNA-PK?

The C-terminal domain of vaccinia C16 has no close structural homologue in the human genome and we indicate this in the manuscript.

We searched for the entire PDB database of structures determined experimentally, and we did not find close structural homologs.

We also searched for the structural homologs in the AlphaFold database of computational predictions, and we did not find close structural homologs.

At the level of sequence analysis, C4 and C16 are conserved in some orthopoxviruses, with some viruses encoding only one of the proteins and some both. We have now performed a more detailed analysis of C4/C16 homologues in several viruses, and looking for the conservation of the “acidic plug” and the region of interaction with Ku.

All this information is now part of a new section of Results, a new Figure 7 and a new Supplementary Figure 9.

8. Page 10, para 3. The sentence starting line 3 and finishing “ ... suggesting that each protein might perform some distinct function additional to their common ability to block dsDNA binding to Ku, perhaps mediated by their less strongly conserved N-terminal domains” needs modification. It is an established fact that the N-terminal domain of C16 binds PHD2 and induces a hypoxic response during normoxia (ref 9) and induce a reprogramming of central energy metabolism (Mazzon et al., JGV 2015), whereas C4 does not do this. These studies should be mentioned in this context.

The reviewer is completely right. This has been done.

9. Fig 6c. The measuring bar referred to is absent.

We have fixed this error in the revised version.

Reviewer #3

Rivera-Calzada et al. report the inhibitory effects of vaccinia virus proteins C16 and C4 on DNA-end sensing by host Ku protein in the cytoplasm and thus their ability to block the innate immune response. The authors demonstrate the virus-encoded C16 or C4 protein each alone blocks DNA binding by heterodimeric Ku70/80 (Ku) and have determined cryoEM structures of the C16-Ku complex at near atomic resolution (2.5 – 3.5 Å). The structure reveals the physical interactions between C16 with Ku and how C16 blocks the DNA-binding channel of Ku. Even though a definitive structure of C4-Ku complex cannot be obtained, 2D projections of the C4-Ku complex indicate a similar mechanism of the viral protein and Ku association. The study is well executed, and the results interesting. After addressing a few minor points listed below, the manuscript will be

suitable for publication in Nature Communication.

1. The first report that the C-terminal half of C16 blocks DNA binding by Ku and its effect on attenuating the host innate immune response was published by Peters et al. in 2013 (<https://pubmed.ncbi.nlm.nih.gov/24098118/>). Although the authors cited this paper in the Methods section for preparing C16 protein, Peters' paper should be cited at the very beginning and credited for their critical finding.

The reviewer is completely right and actually this is our mistake as at some point we deleted this reference from the Introduction and added the wrong one.

We have now revised Introduction to describe in more detail the work by Peters et al, as well as other published literature characterising the proteins C4 and C16.

2. Could the authors please show the boundary between the N- and C-terminal domains of C16?

The reviewer is right that this information was somehow obscured in the previous version of the manuscript.

We now include a simple cartoon in Figure 3a indicating the boundaries of each domain. We have also added some extra information in the text about this in the Results section.

All this will help to clarify the boundaries between domains to the readers.

Can the two domains of C16 be safely separated? Is the N-terminal domain of C16 necessary for inhibiting the DNA-binding activity of Ku?

Yes, the two domains can be separated and it has been shown before that the C-terminal domain (residues 157-331) of C16 is sufficient for binding to Ku and inhibit DNA binding to Ku (Peters et al. PLOS Pathogens 2013). We now clarify this in the Introduction.

In addition, we have expressed the C-terminal domain of C16, and we show that it can dimerise in the absence of the N-terminal domain.

We have performed a size-exclusion chromatography experiment of a construct containing only the C16 C-terminal domain (**Supplementary Figure 5e**). This experiment shows that the C16 C-terminal domain forms dimers on its own, and it reinforces that C16 and C4 interact with Ku as a dimer since this is the domain that binds Ku and the most highly conserved between C4 and C16.

Peters et al. showed that the C-terminal domain. (aa 215 – 331) is sufficient and necessary for blocking Ku from DNA binding. But in the manuscript, the authors cited reference #10 and state that dividing C16 between residues 214 and 215 would fragment the C-terminal domain (p. 8).

This statement is not correct.

Peters et al shows that a construct comprising residues 157 to 331, therefore the entirety of the C-terminal domain according to the C16 structure, is sufficient to bind Ku and block DNA binding.

In contrast, they find that residues 215-331 do not bind Ku (see Fig 2 of Peters et al.), and our current structure explains why the shorter c-terminal fragment comprising only 215-331 doesn't bind Ku.

The new panel in Figure 3a and the changes in the revised version will help readers to visualize the boundaries of the two domains, and the implications of this truncation.

3. Is there any evidence that DNA-PKcs, which forms a protein complex with Ku in nucleus is involved in cytoplasmic DNA sensing and innate immunity?

Yes, there are several reports showing that DNA-PKcs is involved in cytoplasmic DNA sensing, and we mention some recent examples in the revised version of the Introduction.

4. On p. 3, the authors refer to each subunit in the C16 dimer as “monomers”. Shouldn’t they be subunits of the dimeric C16

The use of the word monomer is probably not incorrect, but we have changed this anyway to avoid misunderstandings by the readers.

5. When referring to the C-terminal half of C16 and C4, it may be clearer to distinguish them from the full-length protein by labeling them as C16-C and C4-C rather than C16 and C4.

Thanks for the suggestion, which we have done in both text and figures.

6. C16 and Ku both are shown in cartoon or space-filling diagrams in all figures. It would help to show one in cartoon and the other in space-filling, for example, in Fig. 5 when comparing Ku binding to DNA or C16.

We have modified Figure 5a, so that Ku is in cartoon and C16 and DNA is a space-filling model can be better compared with the DNA – Ku structure.

REVIEWERS' COMMENTS

Reviewer #1 (Remarks to the Author):

The authors nicely improved the manuscript and took well into account the corrections asked

Reviewer #2 (Remarks to the Author):

the alterations made are satisfactory and have addressed all the points made

Reviewer #3 (Remarks to the Author):

The authors have answered all questions thoroughly.

Comments from reviewers

Reviewer #1 (Remarks to the Author):

The authors nicely improved the manuscript and took well into account the corrections asked

Reviewer #2 (Remarks to the Author):

the alterations made are satisfactory and have addressed all the points made

Reviewer #3 (Remarks to the Author):

The authors have answered all questions thoroughly.

Response to the reviewers

The three reviewers are satisfied with our changes to the manuscript. The changes to the first version of the manuscript are highlighted in red colour in the Word file we submit now.

We have now focused our new changes in the editorial requests, and these new changes are highlighted in the text in light green colour.